# Causality Compensated Attention for Contextual Biased Visual Recognition

**Ruyang Liu** [1]  **Jingjia Huang** [2]  **Thomas H. Li**[1]  **Ge Li** ✉[1]
[1] School of Electronic and Computer Engineering, Peking University  [2] ByteDance Inc
{ruyang@stu,geli@ece,thomas@}.pku.edu.cn   huangjingjia@bytedance.com

## Abstract

Visual attention does not always capture the essential object representation desired for robust predictions. Attention modules tend to underline not only the target object but also the common co-occurring context that the module thinks helpful in the training. The problem is rooted in the confounding effect of the context leading to incorrect causalities between objects and predictions, which is further exacerbated by visual attention. In this paper, to learn causal object features robust for contextual bias, we propose a novel attention module named Interventional Dual Attention (IDA) for visual recognition. Specifically, IDA adopts two attention layers with multiple sampling intervention, which compensates the attention against the confounder context. Note that our method is model-agnostic and thus can be implemented on various backbones. Extensive experiments show our model obtains significant improvements in classification and detection with lower computation. In particular, we achieve the state-of-the-art results in multi-label classification on MS-COCO and PASCAL-VOC.

## 1 Introduction

The last several years have witnessed the huge success of attention mechanisms in computer vision. The key insight behind different attention mechanisms (Wang et al., 2018; Hu et al., 2018; Woo et al., 2018; Chen et al., 2017; Zhu & Wu, 2021; Dosovitskiy et al., 2020; Liu et al., 2021) is the same, *i.e.*, emphasizing key facts in inputs, while different aspects of information such as feature map and token query are considered. The impressive performances of these works show that attention is proficient in fitting training data and exploiting valuable features.

However, whether the information deemed valuable by attention is always helpful in real application scenarios? In Fig.1(a), the Spatial Class-Aware Attention (SCA) considers the context *dining table* as the cause of *spoon*, which helps the model make correct predictions even if the stressed region is wrong, but it fails in cases where the object is absent in the familiar context. Fig.1(b) illustrates the problem in a quantitative view: we calculate the mIOU between the attention map (or the activation map for baseline) and the ground-truth mask. As is shown in the figure, although the model with attention gains some improvement measured by common evaluation metrics, the mIOU does not see progress, which means attention does not capture the more accurate regions of targets than the baseline. Moreover, Fig.1(c) illustrates the attention mechanism more easily attends to the wrong context regions when the training samples are insufficient. All above evidence reveals that attention mechanisms may not always improve the performance of the model by accessing meaningful factors and could be harmful, especially in the case of out-of-distribution scenarios.

Fortunately, the causality provides us with a theoretical perspective for this problem (Pearl et al., 2000; Neuberg, 2003). The familiar context of an object is a confounder (Yue et al., 2020; Zhang et al., 2020; Wang et al., 2020) confusing the causality between the object and its prediction. Even though the scene of the *dining table* is not the root cause for the *spoon*, the model is fooled into setting up a spurious correlation between them. On the other hand, the problem is tricky because context is naturally biased in the real world, and common datasets annotated by experts (*e.g.*, MS-COCO (Lin et al., 2014)) also suffer from severe contextual bias. As is well known, networks equipped with attention can learn better representations of the datasets. However, the contextual bias in datasets can be also exacerbated when using the attention mechanism.

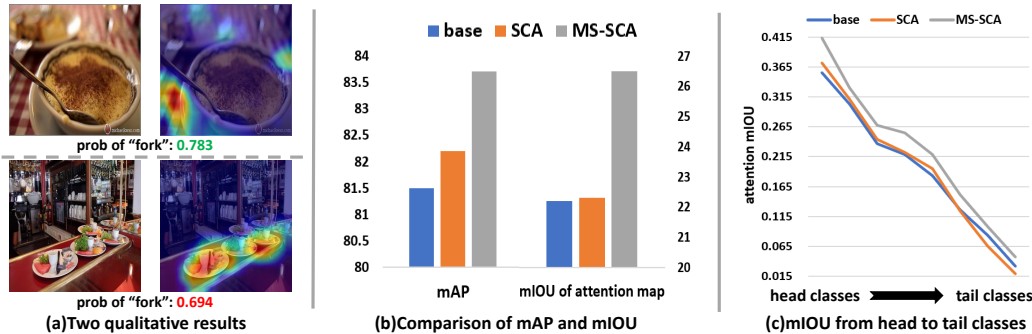

Figure 1: (a) The attention maps of two examples in MS-COCO (Lin et al., 2014) using ResNet101 + SCA (our baseline attention). (b) The mAP and attention map mIOU of ResNet101 baseline, SCA, and the MS-SCA (our de-confounded model). (c) Attention map mIOU of the three models from head classes to tail classes.

To tackle the confounders in visual tasks, a common method is the causal intervention (Pearl et al., 2000; Neuberg, 2003). The interventions in most existing methods (Wang et al., 2021; Yang et al., 2021b; Yue et al., 2020; Zhang et al., 2020) share a maturity pipeline: defining the causality among given elements, implementing the confounders and ultimately implementing the backdoor adjustment (Pearl, 2014). Most of these methods, however, are difficult to implement on attention mechanism and migrate among different tasks. In this paper, we prove that a simple weighted multi-sampling operation on attention can be viewed as the intervention for the attention and confounding context. Based on that, we develop a novel causal attention module: Interventional Dual Attention (IDA). We first employ spatial class-aware attention (SCA) to extract the class-specific information in different positions of the feature map. Then, a multiple-sampling operation with Dot-Product Attention (DPA) re-weighting is implemented upon the SCA, which is essentially the causal intervention and builds a more robust attention map insensitive to the contextual bias. To receive a better trade-off between the performance and computation, we have two versions of IDA: the light one (pure two layers of attention) achieves huge improvements with limited parameter increment; the heavy one (the DPA is extended as a transformer) obtains the state-of-the-arts results with lower computation compared with the popular transformer-based models on the multi-label classification task. Furthermore, improvements in both classification and detection demonstrate the potential of our method to be applied in general visual tasks.

Our main contributions can be summarized as follows:

- We uncover that the attention mechanism may aggravate the existing contextual bias. Qualitative analysis from the viewpoint of causality as well as experimental results is presented to guarantee our arguments.

- We propose an effective and model-agnostic attention module Interventional Dual Attention (IDA). Combining two different layers of attention and the multi-sampling intervention, the module is robust to the contextual bias, and can be extended to the popular transformer to obtain further enhancement.

- We conduct extensive experiments to evaluate the superiority of our method. Rich quantitative and ablation results show that our method can bring about significant improvements on datasets MS-COCO (Lin et al., 2014) and PASCAL-VOC (Everingham et al., 2010) for various computer vision tasks with both CNN-based and transformer-based backbones.

## 2 RELATED WORK

**Attention Mechanism.** Attention mechanism (Itti et al., 1998; Rensink, 2000; Corbetta & Shulman, 2002) aims at imitating the perception system of humans, which adopts sequence signals and selects to focus on salient parts. Hence, no matter in what forms, attention is expected to bias the weight towards the most informative parts of an input signal, and the signal can be varied from a sequence, feature map, or token queries in different tasks.

Over the past several years, attention mechanism has won huge success in a wide range of tasks in computer vision. At the early stage, most of the works apply the attention to sequence-based models and tasks (Bluche, 2016; Stollenga et al., 2014). Then, here comes the age of *attention is all your need* (Vaswani et al., 2017), and many classic attention structures in computer vision arise such as SENET (Hu et al., 2018), Non-local (Wang et al., 2018) and CBAM (Woo et al., 2018). Finally is the popularity of self-attention (Dosovitskiy et al., 2020; Liu et al., 2021; Yuan et al., 2021) in these years, and it can be concluded as the operation among the query, key and value, which even totally replaces the CNN in pure visual tasks. However, as mentioned above, the usage of attention is usually not explicitly supervised. Consequently, the weight of attention is easily biased towards the bias in the dataset.

**Causalities in Computer Vision.** Causality is one of the most important research areas in machine learning including causal discovery (Yehezkel Rohekar et al., 2021; Ng et al., 2021), causal structure learning (Kivva et al., 2021; Akbari et al., 2021) and causal inference (Zhang et al., 2021; Kaddour et al., 2021). Recent years have also witnessed growing applications of causalities in visual tasks such as long-tail classification (Tang et al., 2020; Zhu et al., 2022), few/zero-shot learning (Yue et al., 2020; 2021), and Visual Question Answering (Chen et al., 2020; Niu et al., 2021; Yang et al., 2021a). Significantly, many of these works are about conquering the contextual bias in their tasks. Yue et al. (2020) considered pre-training information as the culprit for the bias and debiased by implementing the backdoor adjustment. Niu et al. (2021) clarified the "good" and "bad" language context and eliminated the bias by pursuing counterfactual total causal effect.

Most similar to our work, Yang et al. (2021b) and Wang et al. (2021) also visited the issue of attention aggravating the bias. Yang dealt with the confounding effect in vision-language tasks, and Wang resolved to build a more robust attention in OOD settings. However, our IDA is inherently different, especially with Wang et al. (2021) in two respects: 1) Different methods: we steer by the implementation of the confounder and other components when approximating the intervention, while Wang et al. (2021) focusing on the partition and annotation of each confounder explicitly. 2) Different applications: Wang et al. (2021) was aimed at OOD settings, which is a direct task for contextual debiasing, but lacks flexibility due to the requirement of pre-defining or clustering each context. In contrast, our model does not have as many limitations, and thus has the potential to be applied in more areas. Furthermore, Wang et al. (2021) has two defects: being evaluated in limited foregrounds and backgrounds (only *animals* in *10* contexts in NICO) and performing poorly when there are multiple objects. The two points are particularly attended to in our method because our model works well in full large-scale datasets of multi-label classification and detection.

## 3   Preliminaries: Causalities in Contextual Bias

In this section, we first demonstrate a causal view of contextual bias in visual recognition. As is conveyed in Fig.2(a), we set up a simple Structural Causal Model (SCM), which clarifies the causality among the context ($C$), image content ($X$) and the prediction ($Y$). In SCM, each link denotes causalities between two entities, *e.g.*, $X \to Y$ means effect $Y$ is generated from $X$. Although the appearance of SCM describing the contextual bias may be different in different tasks (Yue et al., 2020; Zhang et al., 2020; Yang et al., 2021b; Wang et al., 2021), their essence is depicted in Fig.2(a): an extra element ($C$) respectively points to a pair of the cause ($X$) and effect ($Y$) where we want to build a mapping. Next, we will explore the rationale behind the SCM in detail.

$X \to Y$ denotes the predictions depend on the content in images, which is the desired causal effect: to learn a network mapping an image to its prediction. The prediction is unbiased if the target in $X$ is the only causal relation between $X$ and $Y$. $C \to X$ means the context prior determines how an image is constructed by the contents. For example, if all *spoon*s in a dataset appear on the *dining table*, the *dining table* may be regarded as a necessary context when picturing the image of *spoon*. $C \to Y$ exists because the contextual information has a considerable impact on the prediction, *i.e.*, the object itself and its context both affect the recognition of it. $X \leftarrow C \to Y$ together gives rise to a confounding effect: the network will be fooled into building a spurious causal relation *e.g.*, *dining table* is taken as the cause for the prediction of *spoon*. Fig.2(b) further illustrates the role of attention. The attention module can not identify the confounding effect, but focus on intensifying the causal effect $X \to Y$ even if this causal link is wrong, causing the deterioration of context bias.

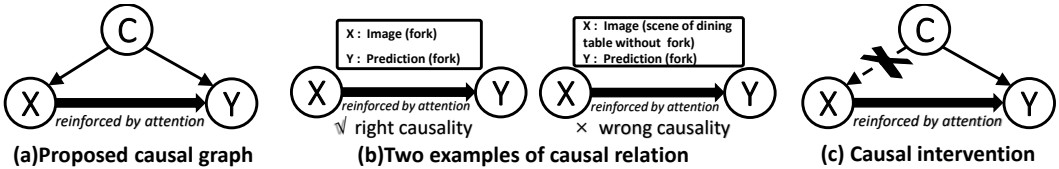

Figure 2: (a) The proposed causal graph for the causalities in contextual bias. (b) Two concrete causal examples. (c) The schematic diagram of causal intervention.

The sole way to eliminate the confounder is by causal intervention. The operation preserves the unavoidable context prediction $C \to Y$ and cuts off the bad link that an object relies on certain context. Common approaches to realizing intervention include RCT, frontdoor adjustment and backdoor adjustment (Pearl, 2014), while backdoor adjustment is the most frequently used in computer vision:

$$P(Y|\mathrm{do}(X)) = \sum_c P(Y|X, C = c)P(C = c), \tag{1}$$

where the do-operation denotes the causal intervention cutting off the edge $C \to X$ as illustrated in Fig.2(c). Free from the interference of confounding path, the network can always learn the unbiased causality between $X$ and $Y$. Here, $X$ can be extended as attention features. Consequently, the reinforcement from the attention mechanism can lead to a better prediction.

## 4 METHODOLOGY

In this section, we present a novel framework that strengthens the robustness of attention on contextual bias. The overview of our model is illustrated in Fig.3. We first introduce our baseline attention: Spatial Class-Aware Attention (SCA), which can obtain class-specific representations but need guidance (Section 4.1). Then, to guarantee the attention module emphasizes the proper causality, we deduce and migrate the backdoor adjustment for the attention, where the intervention is approximated as the multiple sampling with re-weighting upon SCA (Section 4.2). Finally, we present the concrete implementation in our method, *i.e.,* implementing three versions of multiple sampling on SCA (MS-SCA) and implementing the re-weighting as Dot-Product Attention (DPA) or the transformer (Section 4.3).

### 4.1 CLASS-AWARE LAYER

The target of our Spatial Class-Aware Attention (SCA) is to bias the spatial representations toward where the objects are the most likely to show up for each category. Given an image, we can obtain its feature map $\mathbf{X} \in \mathbb{R}^{H \times W \times D}$ from either a CNN-based or a transformer-based backbone, where $D, H, W$ denotes the channel dimension, height, and width. Our purpose is to turn $\mathbf{X}$ into a category-aware representations $\mathbf{E} = \{\mathbf{e}_k\}_{k=1}^{K} \in \mathbb{R}^{K \times D}$, where $K$ is the number of the classes. For each specific class $k$, its representation $\mathbf{e}_k$ is computed from the weighted average of the spatial feature in $\mathbf{X}$. Then, the feature for each category can be reformed according to its unique spatial information:

$$\mathbf{e}_k = \sum_{i=1}^{H} \sum_{j=1}^{W} P(Y = k|X = x_{i,j})x_{i,j}, \tag{2}$$

where $P(Y = k|X = x_{i,j})$ is obtained by feeding $x_{i,j}$ into a linear classifier $\mathrm{f}^{\mathrm{clf}}(.)$ followed with a $\mathrm{softmax}(.)$ regularization.

We adopt SCA as our baseline attention for two reasons: 1) SCA is useful for multi-object tasks. 2) SCA is easier to be affected by contextual bias (Zhu et al., 2017a; Ye et al., 2020; Zhao et al., 2021). Quite a few works adopt similar class-specific representations, due to the insight and interpretability that they empower the models to capture object-aware features in different areas of different images, which is significant for multi-instance tasks. However, pure SCA works badly because it needs guidance to capture causal locations. SCA is designed to underline crucial positions for each class, while the "crucial position" could be the familiar background due to the bias in the dataset. To tackle this problem, other works mainly employ complicated structures (*e.g.*, GCN or Transformer) to further process the representations. By contrast, we argue that a simple intervention (Sec.4.2) is enough to inspire the potentiality of class-aware attention. In the appendix A.4, we show our framework also gains improvement on other classic attention structures.

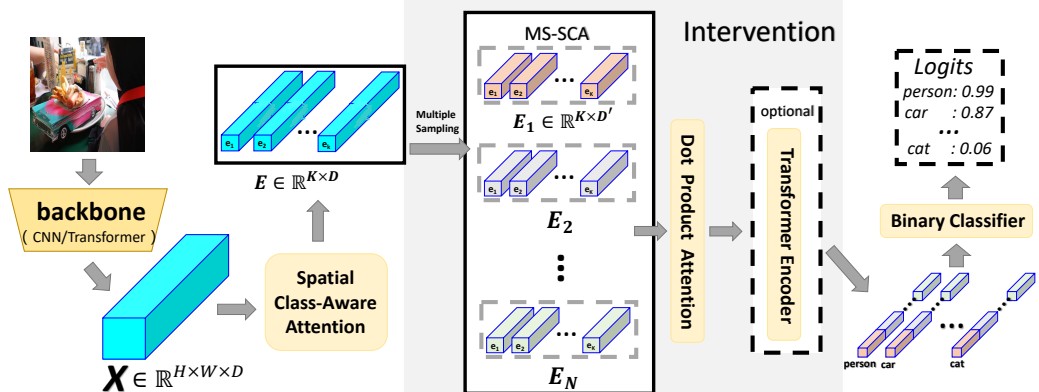

Figure 3: Overview of our proposed model. $X$ could be either image feature from visual backbone or ROI feature from detection backbone. The model is composed of the baseline attention (SCA), the multiple sampling on SCA (MS-SCA), and the second attention layer (DPA or transformer).

## 4.2 CAUSAL INTERVENTION

$P(Y|do(X = x))$ calculates the probability of $Y$ when all $X$ turns to $x$, which is infeasible. Thereby, backdoor adjustment indeed uses realistic statistics (without $do$) to attain the effect equivalent to do-operation. However, it is still challenging that we are required to stratify and sample every possible $c$ for rigorous backdoor adjustment in Eq. 1. In practice, it's difficult to quantificat every possible context, and consequently the $P(C = c)$ is not explicitly observable.

Thanks to the perspective of Inverse Probability Weighting (IPW) (Pearl, 2009), which further reforms the adjustment and simplifies the implementation, we can approximate the sampling on $c$ via the sampling on the observed data, *i.e.*, $(k, x)$. Firstly, we rewrite Eq. 1 to get the equivalent formula:

$$P(Y = k|do(X = x)) = \sum_c \frac{P(Y = k, X = x|C = c)P(C = c)}{P(X = x|C = c)}. \tag{3}$$

$$= \sum_c \frac{P(Y = k, X = x, C = c)}{P(X = x|C = c)}, \tag{4}$$

where $1/P(X = x|C = c)$ is the so-called inverse weight. Although it's hard to sample $c$, in Eq. 4, there is only one $(k, x)$ given one $c$, thereby, the number of $c$ that Eq. 4 would encounter equals the number of samples $(k, x)$ we observe. As a result, the observed $P(Y, X, C)$ can be used to approximate $P(Y = k|do(X = x))$, *i.e.*, the essence of IPW lies in "assign the Inverse Weight $1/P(X = x|C = c)$ to every observed $P(Y, X, C)$, and act as though they were drawn from the post-intervention $P(Y = k|do(X = x))$" (Pearl, 2009). Hence, Eq. 4 can be further approximated as:

$$P(Y = k|do(X = x)) \approx \sum_{n=1}^N \frac{P(Y = k, X = x^n, C = c)}{P(X = x^n|C = c)}, \tag{5}$$

which transforms the summation of C into the sampling of X, and N is sampling times. Here, $C$ in the numerator can be omitted following the common practice of IPW. Then, we model terms in the summation as the sigmoid activated classification probability of the class-aware attention features :

$$P(Y = k|do(X = x)) = \sum_{n=1}^N \frac{P(Y = k|X = x^n)P(X = x^n)}{P(X = x^n|C = c)} = \sum_{n=1}^N \frac{\text{Sigmoid}(w_k e_k^n)P(e_k^n)}{P(X = e_k^n|C = c)} \tag{6}$$

where $w_k$ is the classifier weight for class k and $e_k$ is the feature of class k in Sec. 4.1. Meanwhile, the denominator, *i.e.*, the inverse weight, can be the Propensity Score (Austin, 2011) in the classification model following Rubin's theory, where the normalization effect is divided into the treated

class-specific group ($\|w_k\|_2 \cdot \|e_k^n\|_2$) and untreated class-agnostic group ($\gamma \cdot \|e_k^n\|_2$). Moreover, the confounder (context) in our model is countable, thereby the sampled $e_k^n$ is finite and we simplify the $\|e_k^n\|_2$ as 1. Finally, we compute the ultimate intervention effect by assembling them as follows:

$$P(Y = k|\text{do}(X = x)) = \sum_{n=1}^{N} \frac{\text{Sigmoid}(w_k e_k^n)}{\|w_k\|_2 + \gamma} P(e_k^n), \tag{7}$$

where the implementations of $e^n$ and $P(e_k^n)$ are unfolded in the next section. In Sec.5.3, we will show that the SCA with multiple sampling intervention makes up a "$1 + 1 > 2$" effect.

### 4.3 SAMPLING AND RE-WEIGHTING LAYER

In Eq. 7, the multiple sampling on $e_k$ is crucial to the intervention on the attention. Next, given a fixed feature dimension and sampling dimension (*e.g.,* 2048 and 512), we describe several versions of multiple sampling on SCA (MS-SCA): 1) Random sampling: complete randomness is of no mean and unfriendly to backpropagation, hence, we assign random starting points and random intervals for each sample, and the interval for a single sample is fixed. 2) Multi-head (Vaswani et al., 2017): we equally divide the channel into N groups and take each group as a sample. 3) Channel-shuffle: considering the success of PixelShuffle (Shi et al., 2016; Liu et al., 2021), the shuffle on the channel may also be meaningful. In fact, the multi-head operation is a Channel-shuffle with interval 1. Moreover, channel-shuffle can boost the sampling times with different assigned intervals. Interestingly, we will show by experiments that our model is not sensitive to these choices, indicating that the generic multi-sampling behavior is the main reason for the observed improvements.

Finally is the implementation of $P(e_k^n)$. The simplest way is to assign $P(e_k^n)$ as $1/N$, which assumes a uniform prior of each sample. However, the status of each sample is unequal. Different positions in channel-wise dimension attend to different objects (Zhu et al., 2017b), hence, to achieve better class-aware features, it is necessary to bias the weight into the more crucial samples rather than the average. Another alternative is to introduce the learnable re-weight parameters, nonetheless, the learning capacity is limited and it is hard to scale up the model.

In consequence, we adopt another attention to operate the re-weighting. After the multiple sampling, we get sample-specific class-aware feature $\mathbf{E} \in \mathbb{R}^{B \times N \times K \times D'}$, where $B$ denotes the batch size and $D'$ is sampling dimension. Then, we resize $\mathbf{E}$ into the sequence $\mathbf{E}^s \in \mathbb{R}^{N \times (B*K) \times D'}$. Now, we can implement the scaled Dot-Product Attention (Vaswani et al., 2017) (DPA) to re-weight among different samples, for each class-aware representation:

$$\mathbf{E}^{s'} = \text{softmax}(\frac{(W_q \mathbf{E}^s)(W_k \mathbf{E}^s)^T}{\sqrt{D'}})(W_v \mathbf{E}^s), \tag{8}$$

where $W_q, W_k, W_v$ are the linear projection mapping the sequence into a common subspace for similarity measure. Thus far, we have explained our lightweight version of IDA. Our method increases almost no parameter except the shared KQV projection, and it outperforms all other models with comparable computation. Furthermore, to scale up the model, Eq. 8 can be naturally extended as the transformer, providing opportunities to trade-off between computation and performance:

$$\mathbf{E}^{s+1} = \max(0, \mathbf{E}^{s'} W_1 + b_1) W_2 + b_2. \tag{9}$$

Eq. 8 and Eq. 9 can continue iterating to build a stronger class-specific representation, which becomes our heavyweight version of IDA. Finally, the class-aware representations are fed into a binary classifier followed by a sigmoid activation, and the average logit of all samples is taken as the final logit for that class, where the classifier shares the same learnable weight with $f^{\text{clf}}(.)$ in Sec4.1.

## 5 EXPERIMENTS

In this section, we conduct extensive experiments to demonstrate the superiority of the proposed method. We first introduce the general settings. Then, we present our quantitative results of multi-label classification and detection on three widely used datasets: MS-COCO (Lin et al., 2014), VOC2007, and VOC2012 (Everingham et al., 2010). Finally, we perform various ablation studies and analyses to show the effectiveness of different components in our model.

Table 1: Comparison of mAP (%) on the MS-COCO in multi-label classification. L and H mean the light version and the heavy version respectively, and WH denotes the resolution.

| Methods | Resolutions | mAP | All | | Top3 | |
|---|---|---|---|---|---|---|
| | | | CF1 | OF1 | CF1 | OF1 |
| ResNet-101 (He et al., 2016) | 448 * 448 | 81.5 | 76.3 | 80.0 | 73.5 | 76.0 |
| ML-GCN (Chen et al., 2019c) | 448 * 448 | 83.0 | 78.0 | 80.3 | 74.2 | 76.3 |
| MS-CMA (You et al., 2020) | 448 * 448 | 83.8 | 78.4 | 81.0 | **74.9** | 77.1 |
| CSRA (Zhu & Wu, 2021) | 448 * 448 | 83.5 | 77.9 | 80.3 | 74.4 | 76.5 |
| **IDA-R101(L)** | 448 * 448 | 84.3 | 78.5 | **81.1** | 73.6 | 77.3 |
| **IDA-R101(H)** | 448 * 448 | **84.8** | **78.7** | 80.9 | 73.9 | **77.4** |
| SSGRL (Chen et al., 2019b) | 576 * 576 | 83.8 | 76.8 | 79.7 | 72.7 | 76.2 |
| C-Trans (Lanchantin et al., 2021) | 576 * 576 | 85.1 | 79.9 | 81.7 | 76.0 | 77.6 |
| ADD-GCN (Ye et al., 2020) | 576 * 576 | 85.2 | 80.1 | 82.0 | 75.8 | 77.9 |
| CCD (Liu et al., 2022) | 576 * 576 | 85.3 | 80.2 | 82.1 | 76.0 | 77.9 |
| TDRG (Zhao et al., 2021) | 576 * 576 | 86.0 | **80.4** | 82.4 | 76.2 | 78.1 |
| **IDA-R101(L)** | 576 * 576 | 85.5 | 79.8 | 82.3 | 75.2 | 78.1 |
| **IDA-R101(H)** | 576 * 576 | **86.3** | **80.4** | **82.5** | **76.4** | **78.2** |
| Swin-Base (Liu et al., 2021) | 384 * 384 | 88.4 | 82.2 | 84.0 | 77.2 | 79.9 |
| Swin-Large (Liu et al., 2021) | 384 * 384 | 89.3 | 83.6 | 85.6 | 78.1 | 80.8 |
| **IDA-SwinB(H)** | 384 * 384 | 89.3 | 83.7 | 85.1 | 78.0 | 80.4 |
| **IDA-SwinL(H)** | 384 * 384 | **90.3** | **84.7** | **85.9** | **79.0** | **81.1** |

Table 2: Comparison of mAP (%) between our method and the baseline on the MS-COCO and VOC07 in object detection.

| Methods | Detector | MS-COCO | | | VOC07 |
|---|---|---|---|---|---|
| | | mAP@.5 | mAP@.75 | bbox-mAP | mAP@.5 |
| Baseline | Faster-RCNN-ROIAlign-R50 | 58.1 | 40.5 | 37.4 | 80.5 |
| **IDA (L)** | Faster-RCNN-ROIAlign-R50 | **59.3** | **42.0** | **38.7** | **83.8** |
| Baseline | RetinaNet-R50 | 52.5 | 36.6 | 34.3 | 77.8 |
| **IDA (L)** | RetinaNet-R50 | **55.8** | **38.6** | **36.4** | **80.5** |
| Baseline | DeTR-DC5-R101 | 64.7 | 47.7 | 44.9 | - |
| **IDA (L)** | DeTR-DC5-R101 | **65.7** | **49.2** | **46.1** | - |

## 5.1 EXPERIMENTS SETTINGS

**Implementation details.** Unless otherwise stated, we use ResNet101 (He et al., 2016) pre-trained on ImageNet 1k (Deng et al., 2009) as our backbone. For the multiple-sampling module, we adopt channel-shuffle with start point = 0, intervals = 1 & 2, and sampling dimension = 512 (1/4 of the full dimension), and thus we have N = 8. $\gamma$ is set as 1/32. For the heavy version of IDA, we only have 2 layers of transformer and do not implement the multi-head operation on dot-product attention. There is no extra data preprocessing besides the standard data augmentation (Ridnik et al., 2021; Chen et al., 2019c). The multi-label classification model and the detection model are both optimized by the Binary CrossEntropyLoss with sigmoid. We choose Adam as our optimizer with weight decay of $1e-4$ and $(\beta_1, \beta_2) = (0.9, 0.9999)$. The learning rate is $1e-4$ for the batch size of 128 with a 1-cycle policy. All our codes were implemented in Pytorch (Paszke et al., 2017).

**Evaluation metrics.** For multi-label classification, we adopt the mean average precision (mAP) as our main evaluation metric, and overall/per-class F1-measure (OF1/CF1) as supplements. Separate precision and recall are not employed because they are easily affected by the hyper-parameter. For detection, we employ the mAP@.5, mAP@.75 and mAP@[.5, .95] for MS-COCO, and mAP@.5 for Pascal VOC. In Sec. 5.3, we also evaluate our method on contextual bias benchmarks. The result of classification on Pascal VOC can be found in Appendix A.1.

## 5.2 QUANTITATIVE RESULTS.

**Multi-label classification on MS-COCO.** MS-COCO (Lin et al., 2014) is the most popular benchmark for object detection, segmentation and caption, and has also become widely used in multi-label recognition recently. It contains 122,218 images and covers 80 categories with 2.9 for each image on average. Considering the result of multi-label classification is highly related to the resolution and backbone, we perform experiments with different backbones and different resolutions to make the result more persuasive.

As is demonstrated in Table 1, for CNN backbone, we adopt ResNet101 pre-trained on ImageNet 1k with the resolutions of $448 \times 448$ and $576 \times 576$. The results of lightweight and heavyweight ver-

Table 3: Ablation studies on components of our method with different resolutions. DPA means the dot-product attention in the second layer and Trans means extending the DPA as a transformer.

| Components | | | | mAP | |
|---|---|---|---|---|---|
| SCA | Multi-Sampling | DPA | Trans | 448*448 | 576*576 |
| | | | | 81.4 | 82.7 |
| ✓ | | | | 81.9 | 83.0 |
| | ✓ | | | 82.1 | 83.0 |
| ✓ | ✓ | | | 83.6 | 84.8 |
| | ✓ | ✓ | | 82.2 | 83.1 |
| ✓ | ✓ | ✓ | | 84.3 | 85.5 |
| ✓ | ✓ | ✓ | ✓ | **84.8** | **86.3** |

sions are both present for fair comparisons. The first block shows that our light model has an obvious advantage over other methods with comparable computation (*e.g.*, CSRA (Zhu & Wu, 2021)). For heavyweight models, other models universally adopt 3 or more layers of transformer or GNN (*e.g.*, ADDGCN (Ye et al., 2020), C-Trans (Lanchantin et al., 2021) and TDRG (Zhao et al., 2021)), while ours only has two layers but outperforms all of them, confirming the trade-off between computation and performance in our method is more efficient. For transformer backbone, we use the recently popular Swin-Transformer (Liu et al., 2021) pre-trained on ImageNet 22k. The results in the last block convey our improvement. Based on such a strong backbone, it is very difficult to advance the performance, but our method still brings about considerable progress. In Appendix A.2, we further present the comparison of FLOPs and parameters among some of these methods.

**Object Detection on MS-COCO and Pascal VOC.** Our framework receives the feature map, which can also adopt the ROI features in bounding box as input. Therefore, our method can be seamlessly applied to the detection, including RCNN-like, YOLO-like and DETR-like methods whose neck output the feature maps. To validate that our method can gain improvement in a wider range of visual tasks, we also conduct experiments on object detection on the two widely used datasets. We employ the Faster-RCNN (Ren et al., 2015), RetinaNet (Lin et al., 2017), and DETR (Carion et al., 2020) as our basic detector. The IDA is directly implemented on the RoI feature maps from the neck and the original bbox head is reserved as the residual path which is different from the classification. All the code is implemented in mmdetection (Chen et al., 2019a). The results are shown in Table2. We observe considerable improvements from the baseline on all detectors and datasets, revealing the generalization of IDA on other recognition tasks. Whereas, it must be admitted that the enhancement is not as significant as that in classification, because the role of IDA partly overlaps with the location module in the detection model. Hence, our method obtains greater improvements in the one-stage detection whose object regions are not as accurate as those in two-stage detection.

## 5.3 ANALYSIS.

**Effects of different components in IDA.** To evaluate the impact of our two layers of attention and the intervention, we split and reconstruct our model with different ablation components obeying the default setting. Results on MS-COCO with the resolutions of 448 and 576 are both given. For single multiple sampling, we adopt the channel-shuffle with the default setting and $P(e_k^n) = 1/N$. As is illustrated in Table3, pure multi-sampling and pure SCA both get very limited improvement due to their respective limitation. However, when combining them together, the results see significant improvements, which confirms our arguments that the attention fits the dataset but suffers from contextual bias, and the intervention mitigates the bias and activates the attention. All above tendencies are more obvious for the resolution of 576. We also implement the DPA on multiple samples without SCA, only to find the retrogress, indicating that the re-weighting in the second layer is meaningless if there is no attention information in the first layer. Then, the dot-product attention and transformer further enhance the result at the cost of computation. The improvement of heavyweight IDA on resolution 576 is more evident, meaning our heavyweight version has better model capacity. In Appendix 11, we will show our framework can gain improvement on various attention structures.

**Results on contextual bias benchmark.** Contextual bias is widely seen in various visual tasks, meanwhile, there exist benchmarks particularly designed for contextual debiasing (or OOD setting). To directly evaluate the capacity of our method on contextual debiasing, we conduct experiments on two common benchmarks: ImageNet-A (Hendrycks et al., 2021) and MetaShift (Liang & Zou, 2022). For MetaShift, we adopt the standard domain generalization setting with the subset "Cat vs. Dog". For both datasets, we take the setting without human-annotated context, because there is

Table 4: Different implementations of multiple sampling (above) and $P(e_k^n)$ (below).

| Sampling | $P(e_k^n)$ | mAP |
|---|---|---|
| baseline SCA | - | 81.9 |
| Random Sampling | Average | 82.9 |
| Multi-Head (2) | Average | 82.8 |
| Multi-Head (4) | Average | 83.2 |
| Multi-Head (8) | Average | 83.4 |
| Channel-shuffle | Average | **83.6** |
| Channel-shuffle | Average | 83.6 |
| Channel-shuffle | Parameter | 83.9 |
| Channel-shuffle | DPA | **84.3** |

Table 5: Comparison among our method and other biasing method on MetaShift and ImageNet-A.

| Methods | MetaShift: Cat vs Dog | | | | ImageNet-A |
|---|---|---|---|---|---|
| | d=0.44 | d=0.71 | d=1.12 | d=1.43 | |
| ERM | **84.4** | 60.5 | 35.7 | 24.0 | 30.6 |
| CaaM | 81.9 | 63.8 | **39.2** | **36.1** | **35.6** |
| IDA(L) | 83.4 | **64.2** | 39.1 | 33.6 | 35.3 |

Table 6: Comparsion between our method and CaaM on MS-COCO and contextual biased MS-COCO.

| Method | OOD MS-COCO | Original MS-COCO |
|---|---|---|
| Baseline | 50.0 | 81.4 |
| CaaM | 53.7 | 81.0 |
| IDA-L | **58.6** | **84.3** |

no design for utilizing such partitions in our method. We compare our method with the empirical risk minimization (ERM) and IRM-based CaaM (Wang et al., 2021). As is conveyed in Table 5, on MetaShift, we achieve competitive results among the OOD methods. Specifically, ERM performs the best when the shifts are small; our method is best for moderate shifts; CaaM achieves the best when the the shifts are very large. CaaM obtains the best on ImageNet-A, while our methods have comparable result, and ERM performs much worse. It is also worth nothing that our method is more general for other tasks than the ERM-based and IRM-based methods.

Nonetheless, the benchmarks mentioned above are based on the single-label classification, where the context of single image is simple. While scenes in the wild are naturally multi-label, and the foreground of an object could be the context for other objects. Thereby, for fair comparison (because our method has specific design for multi-label scene), we sample and build a contextual biased subset from COCO to verify our effectiveness. Specifically, we first select six classes of worst performance (e.g., toothbrush) on the original test set. Then we rank the top5 co-exist classes with these target classes (e.g., toilet for toothbrush). Finally, we choose target classes arising in no frequently co-exist classes as positive samples, and the familiar context without target classes as negative samples. We sample half of the training set on MSCOCO in this way to form an OOD test set, and train on the rest training set. As illustrated by Table 6, our method gains a much more significant improvement, while CaaM achieves modest advancement on OOD MS-COCO and even regresses under normal setting. The results prove our effectiveness under context bias, especially when the instances are multiple, where the tranditional debiasing method does not do well.

**Implementation of multiple sampling and $P(e_k^n)$.** In Sec. 4.3, we have introduced three multiple-sampling strategies: Random Sampling, Multi-Head, and Channel-Shuffle, as well as three implementations of $P(e_k^n)$: Average, Parameter (learnable weight) and Dot Product Attention. For Random Sampling and Channel-Shuffle, we adopt sampling dimension = 512 and sampling number = 8 (intervals = 1 & 2 for Channel-Shuffle). For Multi-Head, the sampling number is fixed when the sampling dimension is chosen, thus we choose three sampling dimensions 1024, 512 and 256 with head numbers 2, 4 and 8 respectively. As is uncovered by Table 4, all sampling strategies gain considerable improvement on the original attention, while Random Sampling and Multi-Head with head numbers 2 have lower performance relatively. For the former, random sampling may miss some of the channels and the potential is not fully developed; for the latter, the sample number is too little and the intervention is incomplete. Anyhow, the improvements are not so sensitive to these choices when we have adequate intervention, which indicates the generic multi-sampling behavior is the most important. What's more, the more sophisticated modeling of $P(e_k^n)$ gets better results based on the multiple sampling, and our two layers of attention achieve the best.

## 6 CONCLUSIONS

In this work, we propose an adaptive and effective Interventional Dual Attention (IDA) framework for visual recognition. We demonstrate that the attention mechanism may aggravate the contextual bias in visual tasks, which is blamed on the confounder in causal theory. Then, we define the intervention on the spatial class-aware attention (SCA) that guides the attention to reinforce the correct causal relation, where is intervention is implemented as multiple sampling with the dot-product attention re-weighting. Finally, extensive experiments on different datasets see improvement for both classification and detection, outperforming the state-of-the-arts on multi-label classification with less computation and better performance, meanwhile, a variety of ablation analyses demonstrate the effectiveness of different components in our method.

**Acknowledgements.** This work was supported by Shenzhen Fundamental Research Program (GXWD20201231165807007-202008806163656003) and National Natural Science Foundation of China (No. 62172021).

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

Table 9: Comparison of computation between our method and the state-of-the-arts on the MS-COCO. mAP, FLOPs, and parameters are presented.

| Methods | Param. | FLOPs(448) | mAP(448) | mAP(576) |
|---------|--------|------------|----------|----------|
| ResNet101 | 44.7M | 31.4 | 81.5 | 82.7 |
| CSRA (Zhu & Wu, 2021) | 45.5M | 31.7 | 83.5 | - |
| C-Trans (Lanchantin et al., 2021) | 45.0M | 43.3 | - | 85.1 |
| ADDGCN (Ye et al., 2020) | 48.2M | 32.6 | - | 85.2 |
| CCD (Liu et al., 2022) | 48.3M | 32.0 | 84.0 | 85.3 |
| TDRG (Zhao et al., 2021) | 68.3M | 73.7 | 84.6 | 86.0 |
| **IDA(L)** | 45.6M | 31.7 | 84.3 | 85.5 |
| **IDA(H)** | 55.1M | 33.8 | 84.8 | 86.3 |

Table 7: Comparison of mAP (%) between our method and the state-of-the-arts on the VOC07 in multi-label classification. mAP* means the result of pretraining on MS-COCO.

| Methods | mAP | mAP* |
|---------|-----|------|
| ResNet101 | 92.9 | - |
| MLGCN(Chen et al., 2019c) | 94.0 | - |
| ASL(Ridnik et al., 2021) | 94.6 | 95.8 |
| CSRA(Zhu & Wu, 2021) | 94.7 | 96.0 |
| ADDGCN(Ye et al., 2020) | 93.6 | 96.0 |
| TDRG(Zhao et al., 2021) | **95.0** | - |
| **IDA-R101(L)** | 94.5 | 96.1 |
| **IDA-R101(H)** | **95.0** | **96.4** |

Table 8: Comparison of mAP (%) between our method and the state-of-the-arts on the VOC12 in multi-label classification. All results are tested on official evaluation server.

| Methods | mAP | mAP* |
|---------|-----|------|
| ResNet101 | 92.5 | - |
| HCP(Wei et al., 2015) | 90.5 | - |
| RCP(Wang et al., 2016) | 92.2 | - |
| SSGRL(Chen et al., 2019b) | 93.9 | 94.8 |
| CSRA(Zhu & Wu, 2021) | 94.1 | 95.2 |
| ADDGCN(Ye et al., 2020) | - | 95.5 |
| **IDA-R101(L)** | 94.6 | 95.9 |
| **IDA-R101(H)** | **95.0** | **96.3** |

## A  APPENDIX

In Appendix, we will include more quantitative results, ablation analyses, and qualitative results.

### A.1  MULTI-LABEL CLASSIFICATION ON PASCAL VOC.

Smaller but more delicate, PASCAL-VOC (Everingham et al., 2010) is another widely used dataset in detection and multi-label classification. VOC 2007 has 9,963 images in the dataset with 20 categories, and VOC 2012 has 22,531 images with the same categories. We implement both light and heavy versions of IDA obeying the default setting. Note that some works train the model from scratch (Chen et al., 2019c; Zhao et al., 2021), while others load the COCO-pretrained model (Ridnik et al., 2021; Ye et al., 2020). Hence, for fair comparisons, we implement both of them. Table7 reports the results on VOC 2007. It can be seen that IDA outperforms previous methods, even when some of those models have stronger backbones (*e.g.*, ASL (Ridnik et al., 2021)) or resolutions (*e.g.*, ADDGCN (Ye et al., 2020)). The results of VOC 2012 are conveyed in Table8. Different from VOC 2007, results on VOC 2012 are constrained to be tested on the official evaluation server to guarantee fairness. As is shown, our method also outperforms other state-of-the-art methods by a larger margin.

### A.2  THE FLOPS AND PARAMETERS.

In the main body, we claim that the proposed method can obtain performance improvement with low computations. To verify the argument, we choose the most competitive method recently. We compute the FLOPs and parameters of these methods (of our implementation) using ptflops, and compare them with the baseline and our model. The results are presented in Table 9. Among lightweight models, our model achieves the best performance with similar FLOPs and parameters. For heavyweight models (TDRG), our model achieves better results with half of the computation, indicating our method finds a better way to utilize the increment of parameters.

### A.3  DOES IDA ACHIEVE ACCURATE ATTENTION?

In Sec.1, we uncover that pure attention can not obtain precise attention map. To confirm it experimentally and to validate our method achieves a more robust attention map, we calculate the

Table 10: The class-specific (C-S) and image-specific (I-S) attention mIOU (%) of different models.

| Methods | C-S mIOU | I-S mIOU |
|---------|----------|----------|
| baseline | 16.8 | 22.2 |
| SCA | 17.0 | 22.3 |
| MS-SCA | 19.2 | 25.1 |
| IDA | **20.5** | **26.5** |

Table 11: IDA on different attention structures. Results of mAP on MS-COCO are presented.

| | SENET | CBAM | Non-local | GCNET | SCA |
|---------|-------|------|-----------|-------|-----|
| Vanilla | 81.6 | 81.8 | 82.0 | 82.4 | 81.9 |
| +IDA(L) | 82.5 | 83.0 | 83.0 | 83.5 | **84.3** |

mIOU between the attention map and the ground truth segmentation mask in MS-COCO, which provides an exact evaluation of whether attention emphasizes the right position. In fact, computing Pseudo-Masks from CAM (Selvaraju et al., 2017) of the classification model is a common approach in Weakly-Supervised Semantic Segmentation (Li et al., 2018). For the baseline, we use CAM from the last layers before the classifier as mask, for bare SCA, we use the attention map, and for interventional SCA, we use the average (Multi-head sampling SCA with $P(e_k^n) = 1/N$ ) and the weighted average (IDA) of the attention map from different heads. We compute both image-specific and class-specific mIOU to evaluate the model in different aspects. As is reported in Table 10, pure attention brings about very limited improvement in the attention IOU compared with the baseline, indicating the attention does not help the model find a better location of instances. While our full model outperforms the pure attention and the baseline in both metrics, which proves quantitatively that our method obtains a more accurate and robust attention map.

### A.4    IDA FOR OTHER ATTENTION MODULES.

Recent years have witnessed numerous attention structures for various visual tasks. Considering our method has the potential to be flexibly migrated to different attention modules, we also implement the IDA on four typical visual attention: Non-local, SENET, GCNET, and CBAM on COCO classification. As is shown in Table 11, IDA can give rise to improvement in different attention modules, though the SCA sees the best result and the most progress, because the SCA is the most applicable for the multi-target tasks, whereas it is also the most likely to be affected by the confounding context.

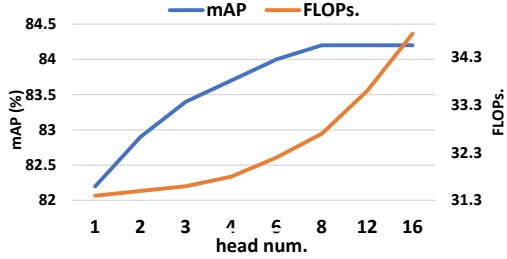

Figure 4: The influence of head number in IDA. We use the IDA with multi-head sampling .

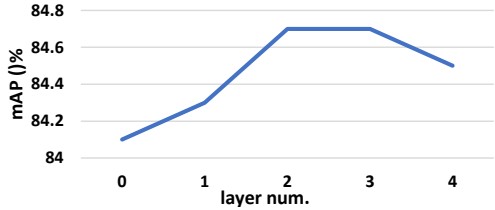

Figure 5: Ablation studies on the number of transformer layers in IDA (H).

### A.5    EFFECTS OF SAMPLING NUMBER.

A larger sampling dimension and more sampling numbers mean more fine-grained sampling and better results, however, the computation also rises up rapidly for self-attention when the sequence is longer. Hence, it is important to carefully choose a rational sampling number. In Fig.4, we study the impact of sampling numbers. For fair comparison, we adopt the multi-head sampling, and project different head dimensions (of different head numbers) into the same 512. As is shown, more sampling heads bring about more improvement, especially in the early stage, where one more sampling means a much more fine-grained approximation for the intervention. The improvement converges when head numbers become large, while the computation keeps rising. As a result, 4-8 heads achieve better trade-offs between performance and cost.

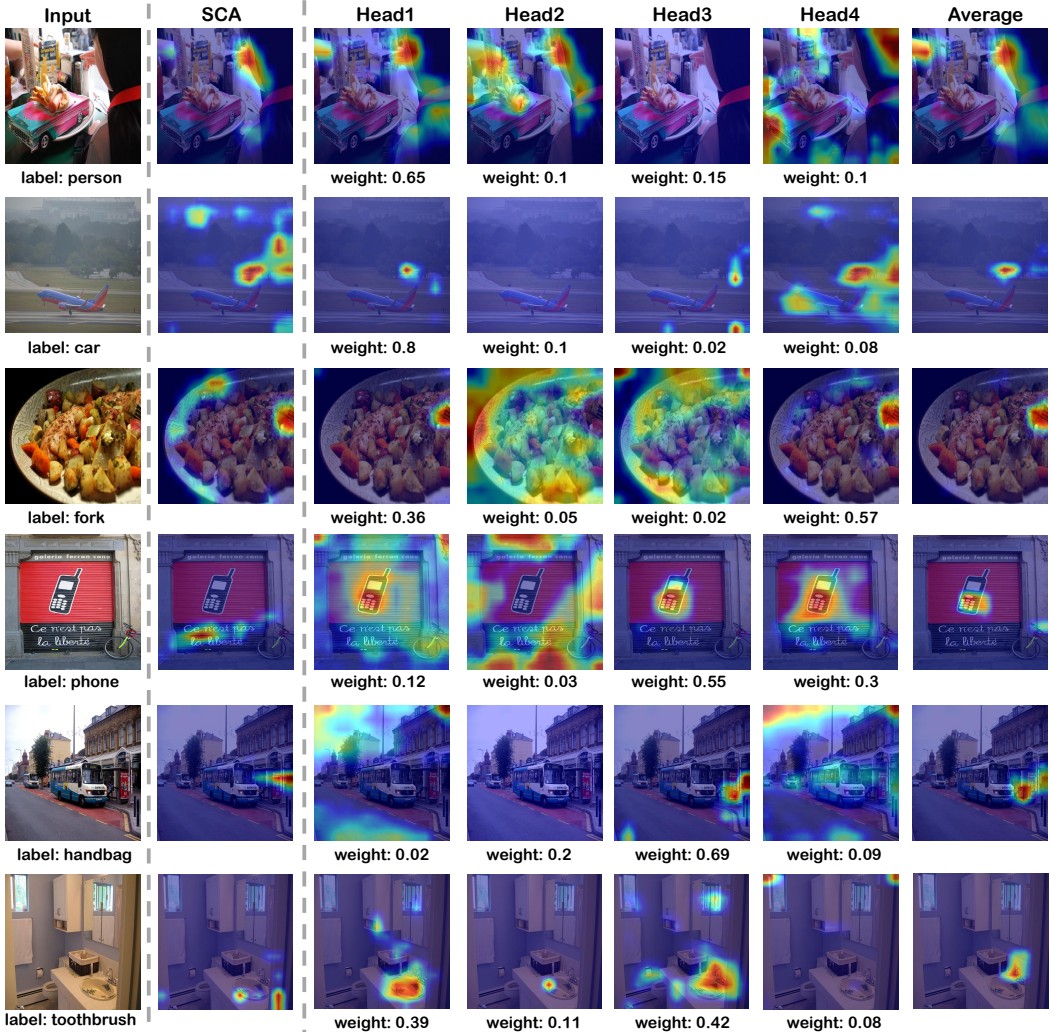

Figure 6: Visualization of baseline SCA and our IDA (L) using CAM, which comes from the spatial attention weight of each class. For our method, we adopt multi-head sampling and use the attention map in different sampling heads and their weighted average, whose weight comes from DPA in the second layer.

## A.6 DEEPER, BETTER?

In theory, the transformer extended from DPA can iterate for many rounds to get a larger model capacity. Hence, we show results on MS-COCO with different layers of transformer for our IDA (H). As is shown in Fig.5, the performance improves when we add the FFN and another layer of transformer for IDA (L), but sees invariability or drop when the layer continues to increase. The possible reason may lie in the function of the second layer that aims at re-weighting the class representations in different samples. Two layers of the transformer are enough to fit needed weights for different samples, and adding more transformer layers is of little use.

## A.7 QUALITATIVE RESULTS.

To demonstrate the effect of our method intuitively, we visualize the parameters via GradCAM (Selvaraju et al., 2017). For the baseline SCA, we use the spatial attention maps of each class, and for our method, we adopt the multi-head sampling of 4 heads and visualize the attention maps of SCA in different heads as well as their weighted average. The average weight comes from the result of $\text{softmax}(QK^T/\sqrt{D/N})$ in the DPA of IDA (L). The result is illustrated in Fig.6. In general, the attention in different samples attends to different things: some for the target objects,

some for the whole background and some for the important context. Then, the weight from the second layer assembles the results, and the weight value is rational as shown. The comparisons between the baseline and our final result further reveal our effectiveness. For high-frequency classes (*e.g., person and car*), our method can locate the fine-grained or hard instances precisely. For the classes easily misled by the context (*e.g., fork and phone*), our attention still finds the target exactly and is not biased to the familiar context. Finally, for the rare or hard classes (*e.g., toothbrush and handbag*), our attention map is inevitably biased more to the context. Compared with the baseline, however, our method can still include the target in the active locations. In the future, we resolve to develop a more fine-grained method to achieve accurate activation maps for these hard instances. Moreover, the context is not always bad for the predictions, *e.g.*, when the training and test set have the same bias distributions. Therefore, it may be better to develop a self-adapting method to judge if the task needs to mitigate contextual bias. As is shown in the visualization, different samples in our method indeed attend to different information, creating possibilities for the above assumptions, which will be studied in our future work.

