# OpenReview forum: "Causality Compensated Attention for Contextual Biased Visual Recognition"
_ICLR.cc/2023/Conference — ICLR 2023 poster_

### Official Review · Reviewer_JL6E · 2022-10-22

**Confidence:** 3
**Correctness:** 3
**Technical Novelty And Significance:** 3
**Empirical Novelty And Significance:** 3
**Recommendation:** 6

**Clarity, Quality, Novelty And Reproducibility:**

(1) Overall, this paper is well written. However, the experiment section could be improved by bringing more relevant analysis to the main paper, and moving the ablation to the appendix.

(2) Due to the lack of comparison with related methods, the paper fails to fully convince me about its novelty and technical contribution.


**Strength And Weaknesses:**

As far as I am concerned, the paper has the following strengths:

(1) Developing visual systems that make decisions by faithfully gathering key information instead of relying on spurious correlations is an important topic.

(2) It provides an extensive ablation study on different components of the method, which facilitates understanding of their contributions.

(3) The proposed method is applicable to a wide range of deep networks, and consistently improves their performance.

However, there are also notable weaknesses:

(1) It is not new to leverage the attention mechanism for addressing issues related to the causal intervention (Yang 2021b and Wang 2021). The paper does not provide a comparison with these closely related studies, which makes it difficult to understand its contribution. I am aware that the paper focuses on a different task (multi-label classification vs standard image classification). However, without a fair comparison, the experiments still fall short of demonstrating the advantages of the method. I would suggest that the authors consider following experimental paradigms in related studies. Note that Wang 2021 not only experiments with NICO, but also other datasets such as ImageNet-A (which makes the statement in Section 2 inaccurate).

(2) Several studies [ref1, ref2, ref3] (see references below) point out that the attention mechanism may not be a good indicator of regions of interest for decision-making. As this paper emphasizes leveraging attention to address biases, I believe it would be interesting to investigate how attention truly benefits the reduction of biases. For instance, what will be the correlation between the permutation of attention and the change of prediction distributions (could be measured by JSD, see details in [ref1])?

(3) Table 1 shows that the proposed method only achieves limited improvements over the compared methods, despite that all of them do not take into account causal relationships. In addition, these results are for standard evaluation and do not offer too many insights in terms of contextual biases. It would be good to move the results in the appendix (e.g., Table 9) to the main paper, which are more relevant to the overall goal.

(4) Looking at the results in Table 9, it appears that the IoU scores are relatively low (i.e., around 0.2), does it imply that the model still largely relies on contextual biases?

(5) In terms of the qualitative results shown in Figure 6, attention maps for most of the heads seem to attend to context while only a few (with higher weights) have a stronger focus on the correct objects. Any idea why? Is the observations somewhat connected to the redundancy of multi-head attention [ref4] or their diverse behaviors [ref5, ref6]?

(6) (minor) Is there a reference for spatial class-aware attention?

(7) (minor) It appears that the paper slightly exceeds the page limit for ICLR.

References:

[ref1] Attention is not Explanation. NAACL, 2019.

[ref2] Attention is not not Explanation. EMNLP, 2019.

[ref3] Is Attention Interpretable? ACL, 2019.

[ref4] Analyzing Multi-Head Self-Attention: Specialized Heads Do the Heavy Lifting, the Rest Can Be Pruned. ACL, 2019.

[ref5] Revealing the Dark Secrets of BERT. EMNLP, 2019.

[ref6] What Does BERT Learn about the Structure of Language? ACL, 2019.


**Summary Of The Paper:**

This paper focuses on addressing the over-reliance on contextual biases, and proposes a new method based on the attention mechanism. It draws inspiration from the causal intervention, and leverages attention together with feature sampling/shuffling to realize the function. Experimental results show that the proposed method is able to outperform the corresponding baselines without considering causal relationships for multi-label classification.

**Summary Of The Review:**

While this paper proposes an interesting method for tackling the issues of contextual biases, its experiments are somewhat insufficient to demonstrate the advantages of the method. There are also some intriguing results that are closely related to the overall objectives of the reduction of biases, but are not carefully studied in the main paper. I am leaning towards rejecting the paper given its current presentation, but I would consider updating the scores if more in-depth analyses/experiments are provided.

---

> ### Author Response · Authors · 2022-11-10
> **Response to Reviewer JL6E （the first part）**
>
> Thank you very much for your review and the useful suggestions. We address all your concerns point by point in two pages. We appreciate any further feedback you have on our current changes.
> ### Q1: Comparison with related methods
> We apologize for the lack of further analysis between our method and Yang 2021b and Wang 2021, especially Wang 2021. In fact, the difficulty of a fair comparison does not lie in the different tasks or datasets, but in the different designs of the model particularly for the two tasks. Specifically, Wang performs debiasing for OOD tasks, where almost all SOTAs rely on the human-annotated or pre-clustering partitions (*i.e.*, the ground-truth context), hence, their methods have been designed to generate and utilize such partitions, which is absent in our methods. In contrast, our method is designed for multi-instance recognition, which most OOD methods do not do well in.
>
> Anyhow, we still conduct extra experiments on the two tasks to form a fair comparison and reveal our contribution. We implement our methods on ImageNet-A, an OOD dataset with severe bias; and migrate Wang's method on COCO classification. The results are posted below:
>
> | Methods  | MS-COCO | ImageNet-A | ImageNet-A* |
> | -------- | ------- | ---------- | ----------- |
> | Baseline | 81.4    | 33.7       | -           |
> | CaaM     | 81.0    | 35.6       | 38.6        |
> | IDA-L    | 84.3    | 35.3       | -           |
>
>  where the ImageNet-A* means using the ground-truth context partitions. As is illustrated, our method achieves comparable results to Wang's method on ImageNet-A without partitions, which indicates our model also works well in the OOD setting. However, our performance falls behind under full supervision, because our method can not make use of the context partitions, which will be dealt with in future work. In contrast, Wang's method performs badly on multi-label classification, leading to recession on the baseline. This defect is also admitted in Wang's paper, where their CaaM cannot accurately attend to multiple objects, let alone multiple labels. As for Yang's method, we have totally different methods (frontdoor VS backdoor) on totally different tasks (visual-language tasks VS recognition tasks), so it is very difficult to have a fair comparison.
>
> ### Q2: The discussion on the reason for debiasing.
> **Why our method benefits debiasing.** Firstly, there may be some misunderstanding. "The attention mechanism may not be a good indicator" is also the conclusion of our paper, and we do not "leverage attention to address biases", but propose to address the problem that attention may be confused by the biases. As for how our method truly benefits the reduction of biases, we give two explanations. One is from the theory, we denote the bias as the confounder, and implement the casual intervention to alleviate the confounding effect, which is one of our main contributions. Another explanation is more intuitive and is provided in Q5.
>
> **What is the correlation between the permutation of attention and the change of predictions?** It is a rather interesting but open question. After carefully reading the given paper ([ref1], [ref2], [ref3]),  we present our own understanding here. As a whole, we do not agree with the conclusion in [ref1], where they draw conclusions from results on SST and argue that the permutation of attention does not affect the predictions. In fact, the experiments in [ref1] on different datasets convey different or even opposite results, and it is unfair to choose the dataset catering to the hypothesis as the standard. The results in [ref2] and [ref3] are also against the opinions in [ref1]: despite the weight of attention is not quite as relevant to the importance of features, attention can still distinguish the importance of input to some degree, which is at least better than the random. From our point of view, attention, as a part of the network, the absolute value of it does not have higher status than the value of other weights. What matters most is if the prediction is sensitive to the change of the certain weight, which is also verified in [ref2].
> ### Q3: Arrangement of experiments
> **Limit improvement over SOTAs.** We admit that the improvement over SOTAs is modest, however, as revealed in Table. 7, we achieve competitive results with superior computation, *e.g.*, $17\times$ fewer FLOPs increment than TDRG.
>
> **Rearrangement of experiments.** Thanks for the valuable suggestions, and we have rearranged our experiments in the new version of the manuscript.

---

> ### Author Response · Authors · 2022-11-10
> **Response to Reviewer JL6E (the second part)**
>
> ### Q4: The scores of mIOU
> The scores of mIOU do seem low even after the intervention, because the result is computed from the attention map without any supervision. In fact, as for the instance segmentation on MS-COCO of using ResNet-101 backbone, under full supervision, the SOTA mIOU is just around 35, while under weak supervision, the SOTA mIOU is also just twenty-odd. The low mIOU can also be explained by the visualization: despite the more accurate location in our method, there still exist many false negatives and false positives due to lack of supervision. Thereby, the absolute value of mIOU is not so important.
> ### Q5: The discussion on qualitative results
> The redundancy or the diverse behaviors may explain the phenomenon to some degree. However, these conclusions are more in point for Bert, while for computer vision, there is a more simple and intuitive explanation. In the feature map extracted from an image, different channel dimensions naturally attend to different image contents, hence, when sampling and visualizing different heads (*i.e.*, different dimensions of the channel), it is reasonable that most heads pay attention to irrelevant locations, while a few focus on correct targets. Inspired by this, besides the theoretical basis from causal intervention, we come up with another perspective that how our sampling and reweighting method reduces the bias: through multiple sampling (in whatever form), we can obtain a more fine-grained attention feature map of the target objects in multiple candidates, while through the reweighting, we underline the desired sample and obtain a more accurate prediction.
> ### Q6: Other concerns
> **Reference for spatial class-aware attention.** Relative reference has been provided in Sec. 4.1.
>
> **The page limit.** Thanks for the kind reminder. The problem has been fixed in new version.

---

> ### Comment · Reviewer_JL6E · 2022-11-15
> **Response to the Rebuttal**
>
> I thank the authors for the additional experimental results and clarification on several issues. After reading the other reviews and responses, I feel that some of the major concerns are not fully resolved, and decided to keep my original rating. Specifically,
>
> **Evaluation of the robustness against contextual biases.**
> As I mentioned in the original review, it is rather difficult to demonstrate the effectiveness against contextual biases with standard multi-label classification (the concern is also raised by other reviewers, *i.e.,* UmuF and GZYH). While the authors carry out a new experiment on ImageNet-A emphasizing the biases, the inferior performance of the proposed method fails to convince me about its advantages.
>
> **Improving the attention mechanism.**
> The authors clarify that the goal of this study is to improve the robustness of attention against biases. In this case, how would you quantify the improvements? I assume IOU could be an option, but the score is not very satisfying and the rebuttal also argues that it is not very important. The authors also highlight the performance gap caused by strong/weak supervision, which makes me wonder if increasing the attention accuracy could address the bias issue.
>
> **Scope of the paper.**
> I tend to agree with reviewers GZYH and  UmuF that the current scope of the paper (*e.g.,* the focus on multi-label classification) does not well align with its motivation for addressing contextual biases. Working on an experimental setting designated for the bias issue would be of great help.
>
> **Regarding the relationship between feature channels and attention allocation.**
> I am aware of vision studies that elaborate on the association between feature channels and semantics (*e.g.,* [ref7], [ref8], [ref9]). However, I believe we should be careful about drawing the conclusion "separation of channels causes self-attention towards background", especially when there is no supporting evidence. For instance, it is at least reasonable to see if standard self-attention without separating channels into different heads results in the same observation. In addition, if we follow the aforementioned logic, multiple feature channels may work jointly to represent a semantic (even though sometimes a single channel is impactful enough to make a difference [ref9]), and randomly shuffling and grouping them could damage the integrity of semantic information.
>
> **Additional References**
>
> [ref7] Network dissection: Quantifying interpretability of deep visual representations. CVPR, 2017.
>
> [ref8] GAN Dissection: Visualizing and Understanding Generative Adversarial Networks. ICLR, 2019.
>
> [ref9] Understanding the role of individual units in a deep neural network. PNAS, 2020.

---

> > ### Author Response · Authors · 2022-11-20
> > **Further Rebuttal for the Response**
> >
> > Thanks very much for the response. For further concerns, we have proposed a detailed clarification. It is appreciated if you have any further comments on our response.
> > ### Q1: Concern about the scope and the evaluation on contextual biased benchmark.
> > In "Further response to all reviewers", we detailedly clarify why we do not take the contextual bias benchmarks as our main task. Besides, the experiments on the contextual biased menchmark ImageNet-A and MetaShift as well as the sampled contextual-biased multi-label dataset are added in the paper (see Table 5,6).
> >
> > ### Q2: Why multi-label classification benchmark.
> > Different from the long-tail bias, contextual bias is widespread in visual datasets without particular sampling. Even on the seemingly "unbiased" dataset like MSCOCO, the context is also generally biased, *e.g.*, 70%+ knives arise up on the dining table and 70%+ oranges arise together with other fruit. Previous works have also proposed that context bias is one of the bottlenecks for multi-label classification [1,2]. Furthermore, the debiasing for multi-label tasks is more challenging and useful for practical application, because images in nature are multi-label and the foreground of one class is the background of another, while nearly all current OOD benchmark is single-label. It is worth mentioning that SOTA contextual debiasing methods (either ERM-based or IRM based) perform badly on multi-label classification, and details can be found in Table 5,6 as well as the "Further response to all reviewers".
> >
> > ### Q3: Further concerns about mIOU.
> > The role of improving IOU is very intuitive: (1) For one thing, when the bias occurs, the attention focuses more on the context and less on the target, hence the more false positive and false negative and lower IOU. Vice versa, when the IOU is significantly improved, it means there is fewer false positive and false negative, and thus the attention focuses more on the objects despite the bias. (2) How to thoroughly remove the context bias? A useful but difficult solution is to annotate where the true objects are, which is the condition of supervised segmentation or weakly supervised segmentation. In other words, the performance of weakly supervised segmentation is the upper bound of the method without supervision, which has been drawn near with our method, thereby, it means that the attention successfully keeps focusing on the objects. What's more, if it is still not convincing for you, we have added the experiments on contextual biased datasets, which are much more direct indicators for debiasing.
> >
> > [1] Causal Intervention for Weakly-Supervised Semantic Segmentation. NIPS2020
> >
> > [2] Contextual debiasing for visual recognition with causal mechanisms. CVPR2022

---

> > > ### Comment · Reviewer_JL6E · 2022-11-20
> > > **Response to the further rebuttal**
> > >
> > > I appreciate the efforts the authors have placed into the rebuttal. I agree that the OOD setting may not be the optimal option for validating the contextual biases issue. However, I would like to stick to my original argument that a standard multi-label classification task also fails to do the job. There are many important questions that are not fully answered under the current scope of the paper, for instance: (1) How do we know if the improvement in the standard setting  (especially when the scores are not good enough) is indeed a result of the reduction of biases? (2) What is the relationship between attention accuracy and contextual biases? (3) Why resolving contextual biases does not help OOD, which is a setting that is supposed to emphasize more on addressing overfitting to spurious correlation? Perhaps reviewer sj7K's view may more accurately characterize this paper, "it is more re-packaging of attention mechanism to extract class-specific information".
> > >
> > > Nevertheless, I believe quantifying models' reduction of biases itself is a challenging problem, and it could be too harsh to expect a single conference paper to answer so many questions. Therefore, I have decided to raise my score.

---

> > > > ### Author Response · Authors · 2022-12-05
> > > > **Response to the three questions**
> > > >
> > > > Thank you very much for the further response. For the three main concerns, we present our understanding as well as some new results. It is appreciated for any further feedback.
> > > >
> > > > ### How do we know if the improvement in the standard setting is indeed a result of the reduction of biases?
> > > > It is rather difficult to answer this question directly, but we propose to reveal this from a side view. Specifically, we evaluate some of the SOTA methods for multi-label classification on the OOD benchmark, in order to compare the ability of debiasing among different methods that improve the standard setting. We implement three methods representing three directions: ASL, TDRG, and CCD. ASL is the loss-based method free from architecture modification; TDRG has complex structures with tiled GCN and transformers, improving the standard setting by large model capacity and heavy computation; CCD shares the same insights with us to improve the multi-label classification benchmark by contextual debiasing：
> > > >
> > > > | Method | MS-COCO | ImageNet-A | Extra FLOPs | Extra para |
> > > > | ------ | ------- | ---------- | ----------- | ---------- |
> > > > | ASL    | 85.0    | 32.9       | 0           | 0          |
> > > > | TDRG   | 86.0    | 34.0       | 42.3        | 23.6M      |
> > > > | CCD    | 85.3    | 34.9       | 0.6         | 3.6M       |
> > > > | IDA    | 86.3    | 35.3       | 0.4         | 0.9M       |
> > > >
> > > >  As is shown, ASL performs the worst on OOD data (worse than the baseline 33.7), it is perhaps because ASL polarizes the grads which worsen the bias. TDRG brings very limited improvement, revealing blindly lifting the model capacity is of little use for the bias. CCD and our model both bring about more significant advancement on bias benchmark, which indicates the improvement in the standard setting comes more from debiasing than other types of methods, while our method is superior to CCD with the same motivation.
> > > >  ### What is the relationship between attention accuracy and contextual biases?
> > > >  The improvement of contextual bias means the model focus more on the objects themselves. In this paper, we employ segmentation IOU to reveal the attention accuracy, and further demonstrate the debiasing. Despite the improvement, the score of mIOU is very low, which has raised the question that if the model does focus on the objects. To further address this problem, we propose a new metric to evaluate attention accuracy: bbox mIOU. Specifically, we compute the ratio between the attention area in the bounding box and the whole attention area. Compared to the segmentation, the bbox mIOU is rougher but more suitable for our target: focusing more on foregrounds means focusing on the areas of the objects, rather than fitting exactly where the objects are. The result on MS-COCO is shown below:
> > > >
> > > >
> > > > | Method   | BBox IOU |
> > > > | -------- | -------- |
> > > > | Baseline | 75.74    |
> > > > | +SCA     | 76.06    |
> > > > | +IDA     | 82.23    |
> > > >
> > > > As is shown, compared to the segmentation mIOU, IDA has even more improvements for attention accuracy, and is more persuasive for the existence of debiasing. It could be better if we can evaluate on the OOD benchmark, but it is a pity that such benchmarks have no annotations for the detection or segmentation.
> > > >  ### What is the best setting for the reduction of biases?
> > > >  It is a difficult and open question. We present three potential directions from our points of view: (1) Traditional methods for OOD dataset (ERM and IRM) do work for the completely contextual biased data, however, the problem is that they also damage the recognition of IID data. Therefore, a potential solution is to identify if the samples are out-of-distribution, applying the ERM or IRM-based methods for OOD data and normal methods for IID data. (2) Develop new methods for the contextual bias that performs the debiasing without the cost of damaging the IID data, which is also the purpose of our paper. One option is causality, which has been proven effective for contextual debiasing in many fields. (3) The pretraining, especially the large-scale visual-language pretraining. The so-called contextual bias is not a problem when we have extremely abundant knowledge. In fact, the CLIP can achieve a zero-shot accuracy of 70% on ImageNet-A, which is far better than any other method without pretraining. Thereby, employing a well-pretrained model to address the contextual biased data may be a good solution.

---

### Official Review · Reviewer_GZYH · 2022-10-24

**Confidence:** 3
**Correctness:** 3
**Technical Novelty And Significance:** 3
**Empirical Novelty And Significance:** 2
**Recommendation:** 6

**Clarity, Quality, Novelty And Reproducibility:**

Clarity: The paper needs a significant polishing\
Quality: The core idea is good, but current evaluation is limited\
Novelty: I think the proposed framework is novel\
Reproducibility: Code is submitted

**Strength And Weaknesses:**

### Strength

- The motivation in Figure 2 is interesting. Addressing contextual bias via modifying the prediction from $P(Y|X)$ to $P(Y|do(X))$ is clear.
- The proposed method outperforms the baselines for multi-label classification.

### Weakness

**Concerns in presentation**

My major concern is the overclaiming presentation. The proposed method is just an improved Transformer block for multi-label classification. However, I did not catch this point until carefully looking at the method. The current title, abstract, and introduction look like the proposed method can improve the robustness of ViT on the contextual bias, e.g., applied for general classification [1] or even object detection. The paper should clarify its scope.

The method is also overly complex and missing some important details. If I understand correctly, isn't spatial class-aware attention (SCA) just an average pooling of spatial class-wise features (which are widely used [2])? So the baseline SCA is just pooling the class-wise feature and applying a binary classifier for each class vector. Then, the proposed method is just an interaction layer of this class vector: convert a R^{KxD} vector to a refined R^{KxD} vector. Figure 3 and the explanation in Section 4 could be significantly improved. Currently, the overview of the method says the structure to be (1) SCA, (2) MS-SCA, (3) DPA. However, it does not match the title of the subsection, which confuses the readers. Also, neither subsection 4.2 nor 4.3 explains what the MS-SCA is exactly -- one can merely guess that Eq. (7) is MS-SCA.

In addition, the paper has lots of typos:
- bare -> base in Figure 1 caption?
- Sec.7 -> Eq. 7 in page 4?
- number of c Eq. 4 -> number of c in Eq. 4?

**Concerns in scope and evaluation**

Another main concern is the scope and evaluation. The proposed causal framework is general and may be applied beyond multi-label classification. Extending the method for general classification or even object detection would highly strengthen the paper. Also, I encourage the authors to include common benchmarks for measuring contextual bias such as MetaShift [1].

[1] Liang & Zou. MetaShift: A Dataset of Datasets for Evaluating Contextual Distribution Shifts and Training Conflicts. ICLR 2022.\
[2] Yun et al. Re-labeling ImageNet: from Single to Multi-Labels, from Global to Localized Labels. CVPR 2021.

**Summary Of The Paper:**

This paper aims to reduce contextual bias in multi-label classification. To this end, the paper aims to remove the effect of unseen context $C$ from the prediction. Specifically, given a casual structure {$C \to X, C \to Y, X \to Y$}, the usual classifier predicts $P(Y|X)$ which implicitly reflects the unseen $C$. Instead, the paper aims to predict $P(Y|do(X))$ to discard the contextual bias. This probability can be computed by backdoor adjustment (Eq. (1)) and inverse probability weighting (Eq. (4)). This term can be estimated by setting the context $c$ as a pair or class and image $(k,x)$, leading to the final estimator (Eq. (7)). The proposed method shows good performance on various multi-label classification benchmarks.

**Summary Of The Review:**

I think the overall idea is interesting. However, I'm lean toward borderline reject for the current version of the paper. I hope the revised paper includes:
- Polish presentation -- clarify the scope and clearly explain the proposed components (complex $\neq$ novel)
- Extend the scope more than multi-label classification and add empirical validations for those experiments

---

> ### Author Response · Authors · 2022-11-10
> **Response to Reviewer GZYH**
>
> Thanks for the valuable comments. We are sorry for some misunderstandings in the current version of the paper. We believe that the suggestions for improvement provided by your review can easily be accommodated within the revision cycle. we are currently preparing a new version of the manuscript. We have addressed all of your concerns in the rebuttal. It is appreciated if you have any further feedback on our current and later changes.
> ### Q1: The scope of our paper.
> Sorry for the confusion. In the paper, we argue that our causal framework is general for various visual tasks, but are questioned that there are only experiments on multi-label classification. Indeed, we have also presented the experiments on object detection in Table. 6 in the appendix, which may be missed during the review. Besides, we conduct an extra experiment on ImageNet-A, a dataset suffering from severe contextual bias, and the details can be found in the reply for Review 4 JL6E (the section "Q1: Comparison with related methods"). The improvement on three tasks may verify our arguments that our framework is general for classification or even object detection, while the improvement on multi-label classification is the most obvious due to some specific designs for this task.
>
> ### Q2: The polish for presentation.
> We apologize for the rough writing for our method. Indeed, our framework itself is simple: we propose a baseline attention SCA, and perform the multiple sampling and reweighting operation on it. In Section4.2, we spend much content explaining why the intervention can be approximated as the multiple sampling with reweighting, which may be complex and abstruse. In the new version of the manuscript, we have presented a more clear clarification for the proposed components in the overview of Sec. 4 and fixed the typos. We have also updated the figure 3. More elaborate polishment of the figure 3 is underway and will be presented later. Particularly, MS-SCA means the multiple sampling on SCA, which is explained in several places in the new version.
>
> ### Q3: Some misunderstandings.
> We find there may be some misunderstanding and present some detailed explanations here.
>
> **"The proposed method is just an improvement Transformer (or ViT) block."**  In fact, our method is not limited to the self-attention in transformer, but can improve various types of visual attention like SENET, CBAM, and Non-local, which is studied in appendix A.3.
>
> **"SCA is just an average pooling of spatial class-wise features."** Our SCA obtains the spatial class-wise features through the "pooling" of feature map, rather than "the pooling of spatial class-wise features". Besides, the "pooling" is not  average but weighted average, where the weight comes from the pre-classification of the feature map.

---

> ### Comment · Reviewer_GZYH · 2022-11-13
> **Response to the Rebuttal**
>
> Thanks for the rebuttal. However, my initial concerns are not addressed. Thus, I keep my original rating of reject.
>
> **Scope of the paper**
>
> Even after the rebuttal, the main text of the paper only discusses the multi-label classification, which highly limits the Scope of the paper. The paper should include a detailed discussion and empirical evaluation of object detection in the main text.
> - How to apply the proposed framework for object detection? The paper does not discuss details about this. The method figure (Figure 3) only shows the multi-label classification. Can the proposed framework be applied to arbitrary object detection frameworks, including RCNN-like, YOLO-like, DETR-like, and more?
> - The baseline results in the Appendix are far worse than the recently reported results, making the overall results not convincing. See [1] for an example. Can the proposed method be applied to the SOTA object detection methods?
>
> Also, the revised paper does not include the experiments on popular contextual bias benchmarks (even though the paper's title is about this), which validates that the performance gain of the proposed method indeed comes from reducing the bias. An in-depth analysis of contextual bias should be included in the main text. The rebuttal does not provide the results on the MetaShift [2] benchmark that I suggested. The new results on ImageNet-A are not convincing as the proposed method (35.3) is worse than the prior work CaaM (35.6). If the comparison is not fair, why not design an experiment that verifies the effectiveness of the proposed method?
>
> [1] https://paperswithcode.com/sota/object-detection-on-coco \
> [2] Liang & Zou. MetaShift: A Dataset of Datasets for Evaluating Contextual Distribution Shifts and Training Conflicts. ICLR 2022.
>
> **Other comments**
>
> - Presentation: Could you highlight the revised texts for easy comparison? Also, if the paper wants the readers to read the Appendix, it should include a mention in the main text of which part of the Appendix includes which results.
> - Different attention: If the paper wants to claim its universality on different attentions, it could revise Eq. (8) in a more general form and then show ViT as a special case. Though, it is not surprising that the proposed method can be applied to other pairwise attentions.

---

> > ### Author Response · Authors · 2022-11-20
> > **Further Rebuttal for the response**
> >
> > Thanks very much for the response. We have updated our manuscript according to your every suggestion, including the implementation of detection and more experiments for detection and contextual debiasing. It is appreciated if you have any further feedback on our current changes.
> >
> > ### Q1: Concern about detection.
> > We have moved the experiments of detection to the main body. Details about the application of detection are also added in the paper. Our framework can be applied to most of the detection frameworks given the bbox feature map, including RCNN-based, YOLO-based and DETR-based. Furthermore, we add the experiments of our method on DETR (see Table 2), which is much more powerful than the previously proposed one and also sees considerable progress. We are sorry for the gaps compared to the SOTAs on PaperWithCode. However, current detection SOTAs rely on (1) Large-scale image-text pretraining, *e.g.*, GLIP. (2) Huge model scale, *e.g.* SwinV2. (3) Object365 pretraining, *e.g.*, DINO. The computation cost is tremendous (*e.g.*, 200M InternImage needs about 7 days on 32 A100-80G, let alone the 600M or 2G one), which is unaffordable for us.
> >
> > ### Q2: Concern about contextual bias benchmarks.
> > In "Further response to all reviewers", we have clarified the reason why we do not take the contextual bias benchmarks as our main task. The experiments on the contextual bias benchmarks are included in the main paper (Table 5,6). Specifically, we evaluate our method on the ImageNet-A and the suggested MetaShift (they are of the same type dataset..) as well as the sampled contextual-biased multi-label dataset. For the opinion that the inferiority result on ImageNet-A is unconvincing, we have clarified that pursuing the SOTA on contextual bias dataset conflicts with the applying for general tasks, *i.e.*, it is very difficult to achieve SOTA on every task (details can be found in "Further response to all reviewers").

---

> > > ### Comment · Reviewer_GZYH · 2022-11-27
> > > **Response to the further rebuttal**
> > >
> > > Thank you for following my suggestions. I believe the updated results in object detection (Table 2) and contextual bias benchmarks (Tables 5 and 6) would highly strengthen the paper's contribution. I raised my score to be weak accept.

---

### Official Review · Reviewer_sj7K · 2022-10-24

**Confidence:** 3
**Correctness:** 3
**Technical Novelty And Significance:** 3
**Empirical Novelty And Significance:** 3
**Recommendation:** 6

**Clarity, Quality, Novelty And Reproducibility:**

The paper is well-written and easy to follow. It has given a background on what is the problem in most of the attention-driven approaches for visual recognition and justification on how the proposed approach advances in this direction.

There is some element of novelty but it is unclear to me. I have raised this in the weakness section. It would be nice if the authors clarify this in rebuttal and I will reconsider my decision.

The quality is good. Experimental evaluations are carried out to justify the proposed approach. In the weakness, I have mentioned about the use of heavier backbones. Why? why not light-weight model with standard resolution of 224.

The paper explains well about the implementation. The source code is attached as supplementary and the authors have promised to release the source code. Thus, there is little doubt on reproducibility.

**Strength And Weaknesses:**

Strengths:
The paper explained the problem  of contextual biases in attention models for solving visual recognition problems.

To address this the proposed interventional dual attention (IDA) consisting attention and multi-sampling intervention.

Evaluation of the proposed approach in well-known datasets COCO and VOC, and comparison to the state-of-the-art.

Ablation study justifying the components IDA and visualization using GRAD-CAM to show the attention map.

Weakness:
I have doubt about the novelty aspect of the approach. Although the paper claims it is novel but it is more re-packaging of attention mechanism to extract class-specific information. The causal intervention seems to be incremental. The final eqn 7 is more of standard weighted attention?

The approach is evaluated using heavy backbones (ResNet-101 or Swin-Base) with image resolution of 384, 448 or 576. Is there any justification to use heavier backbones or large resolution? Experiments should have been carried out with lighter model ResNet-50 or NAS-Net with input resolution of 224 which is widely used for various visual recognition problem.

The claim “attention does not capture the more accurate regions of targets than the baseline.” This could be for a specific image example. Generalizing this claim requires thorough examination which is not done in the evaluation process.

The spatial class-aware attention is sensitive to spatial augmentations?  The impact of spatial augmentation (scale, crop, rotation, etc.) on SCA is not discussed.

I feel there are some recent work on spatial attention by exploring semantic regions for fine-grained visual recognition such as CAP (Behera et al., AAAI 2021), SR-GNN (Bera et al., TIP 2022) exploring GNN and Wharton et al. (BMVC 2021) exploring relationships between hierarchical structures via GCN and could be cited.

Where is the table containing FLOPS and parameters, and comparison to the baselines and state-of-the-art models? In supplementary, it says Table A.2 which is pointing to the Table 8 that says M

**Summary Of The Paper:**

The  article propose a method to learn casual image features for contextual bias via an attention mechanism called interventional dual attention (IDA). The mechanism consists of two attention layer with multiple sampling intervention that protects attention from cofounder context. The approach is evaluated on well-known datasets (COCO, VOC) and achieves SotA performance on multi-label classification.

**Summary Of The Review:**

The motivation of proposing interventional dual attention for visual recognition is very good but the novelty aspect on the proposed mechanism is not fully explored. The proposed approach is thoroughly evaluated and compared to the state-of-the-art (SotA). It lacks the description on the computational complexity and critical discussion on performance of the model in comparison to the SotA. Please refer to the sections 2/3 for the details. This is a good work I would like to see authors rebuttal to my posed questions.

---

> ### Author Response · Authors · 2022-11-10
> **Response to Reviewer sj7K**
>
>
> Thank you for your comments. We appreciated you giving positive feedback on our paper. All your concerns are addressed point by point below. It is appreciated if you have any further comments on our response.
> ### Q1: Doubt of novelty.
> **Re-packaging of attention mechanism and causal intervention seems incremental.** We propose to improve existing attention for the robustness on the contextual bias, which can be viewed as "re-packaging" in some sense. However, the causal intervention is not incremental or dispensable. As is revealed in ablation study (Table 4) and further analysis (Table 6), the baseline attention SCA just achieves very limited improvements on bare ResNet101, seeing only 0.5% and 0.2% on mAP and mIOU. It is the causal intervention (*i.e.*, multiple sampling with reweighting) that brings the most promotion.
>
> **Final Eq. 7 is more of standard weighted attention?** Eq. 7 itself is not an attention. Instead, it takes the output of the attention (the attentioned feature $e_k$) as input, and conducts the multiple sampling and reweighting operations on the attention feature.
>
> I guess you may doubt that these operations seem like the standard multi-head self-attention in transformer? For this problem, we have three clarifications: **(1)** The multi-head and self-attention are just the subset in our method, (*i.e.*, setting the multiple sampling as multi-head and the reweighting as self-attention). In Table 5, we have shown other options can obtain similar or even better results. **(2)** How the multi-head and self-attention are formed is totally different. In transformer, the multi-head operation is conducted on self-attention itself. In contrast, we conduct multi-head on the baseline attention (SCA or other attention in Table 10) in the first layer, and adopt self-attention as reweighting in the second layer, *i.e.*, the input sequence for self-attention is reshaped from the multi-head output, rather than the patch tokens in ViT. **(3)** In fact, some previous works (*e.g.,* Interventional Few-shot Learning) have implied that the multi-head operation can be viewed as a form of intervention. However, we are the first one to present the complete deduction as shown in Sec 4.2. We prove that it is the general multiple sampling operation that really works, and the reweighting is also important, which is absent in previous work.
>
> ### Q2: Why heavy backbones with large resolution?
> We adopt heavy backbone with large resolution for two reasons: **(1)** Current SOTA models of multi-label classification all use ResNet101 with resolution 448 or 576 as baseline. Thereby, for a fair comparison, we keep pace with them. **(2)** Baseline with larger backbones or resolutions is harder to improve. Small backbone or resolution may suffer from the limited model capacity or incomplete feature extraction, thus is easier to advance. While for the heavy one, simple techniques like enlarging the model capacity don't work. Especially, we implement our method on ResNet-50 with resolution 224 and obtain mAP of 72.9 on the baseline 67.8, achieving +5.1% compared to +3.3% in ResNet-101 with resolution 448.
> ### Q3: The claim "attention does not capture the more accurate regions".
> Besides the visualization, there are two points standing this claim. **(1)** In Table 6 (Table 9 in the original version), we presented the mIOU of different components to evaluate if they achieve an accurate attention map. As is conveyed, the baseline attention SCA sees little improvement on the bare ResNet101, which at least testifies SCA does not capture a more accurate region. **(2)** This conclusion is also drawn by many other works, besides the two papers mentioned in the last paragraph of Sec. 2, Reviewer JL6E also presented several papers in NLP field containing similar viewpoints.
> ### Q4: Impact of spatial augmentation.
> Thanks for the reminder. Indeed, these augmentations are included in the baseline following previous works. We conduct an extra experiment to check the impact of these spatial augmentations.
>
> |          | Bare | Scale224 | Cutout0.5 | Rotation |
> | -------- | ---- | -------- | --------- | -------- |
> | Baseline | 78.3 | 76.0     | 80.2      | 79.5     |
> | IDA-H    | 82.0 | 79.9     | 83.7      | 82.9     |
>
> As is illustrated, as a whole, our method achieves larger improvement on a weaker baseline, while the gap is negligible, which demonstrates our method is not so sensitive to spatial augmentations. Moreover, the results may have the same explanation in Q2: a stronger baseline is harder to improve.
> ### Q5: More related works.
> Thanks for the suggestion. These works will be mentioned in the new version of the manuscript.
> ### Q6: Where is the FLOPs.
> Sorry for your confusion. FLOPS and parameters are in Table 7 in the original paper. We have fixed the labeling flaws in the paper. Now, the data is in Table 9 in the new version.

---

> > ### Comment · Reviewer_sj7K · 2022-11-16
> > **Response to the Rebuttal**
> >
> > Thank you authors for the rebuttal. Most of my comments are addressed but the novelty aspect is not convincing. I have gone through the reviews from other reviewers, as well as the respective rebuttals. Thus, I am keeping my original decision of "marginally above the acceptance threshold".

---

> > > ### Author Response · Authors · 2022-12-05
> > > **Thanks for the response**
> > >
> > > Thanks again for your comments.
> > >
> > > We have carefully dealt with all other reviewers' points. We hope that some of these responses can also approach your concerns. We'd be happy to address any remaining points. If our response is satisfying, we kindly ask you to consider raising the score. Thanks so much!
> > >
> > > With sincere regards,
> > >
> > > Authors of Paper 1897

---

### Official Review · Reviewer_UmuF · 2022-11-02

**Confidence:** 2
**Correctness:** 2
**Technical Novelty And Significance:** 2
**Empirical Novelty And Significance:** 2
**Recommendation:** 3

**Clarity, Quality, Novelty And Reproducibility:**

As described above, the paper is not at all clear when it comes to connecting motivation to method and to results. The empirical results show modest improvements over the baselines, but these may very well be the result of additional model capacity rather than some form of causal reasoning as advertised.

**Strength And Weaknesses:**

I struggled to understand the approach and in particular to connect the stated motivation with the method being proposed as well as the results. The paper starts by describing an often observed problem, namely that common recognition methods struggle when the combination of foreground and context violates expectations. Based on the presentation, it is neither clear how the proposed method would address this, nor why the (small) improvements over the baseline indicate that the problem is indeed being successfully addressed.

The presentation of "causalities in contextual bias" in section 3 is confusing. The presented structural causal models (see Fig 2) only model the problem at a very abstract level. Its also for example not clear what is being proposed in (2a), if the directions of the edges make sense, why a misleading relationship between X & Y might be reinforced specifically by attention, and what it means to break the connection between the context and the image (2c) both conceptually and in practice. The connection between what is presented here and the method is very tenuous. How is the "multiple-sampling operation with Dot-Product Attention (DPA) re-weighting" "essentially [a] causal intervention"? What happens in the case when the context is necessary to correctly identify e.g. a small/low-resolution object?

The paper argues that an attention map that tracks more closely with the ground truth mask of the object is a sign of the method's success. Such results are presented e.g. in Fig 1b and in the Appendix (A.4, see final paragraph in section 5). These results show modest improvements and are also based on the faulty assumption that the context is to be ignored lest it be faulty, when in fact one would rather prefer a model that can take context into account when needed.

**Summary Of The Paper:**

This paper proposes an attention mechanism that is more robust to misleading visual context. The motivation follows from the fact that current recognition models tend to mistake correlations (e.g. between background and objects) for causal factors, leading to mistaken predictions when e.g. observing the background without the foreground object.

The proposed attention mechanism comes in two variants: one that relies on two successive attention layers, and a more heavyweight version which involves a transformer-based encoder inserted between the aforementioned layer. These extra processing layers are combined with a form of sampling-based intervention, which should allow the model to ignore spurious context.

The method is evaluated on two multi-label classification tasks (MS-COCO and Pascal VOC), and compares favourably to competing methods. An ablation study shows how different components of the approach improve over the baseline.

**Summary Of The Review:**

The paper presents a method that to me is only tenuously connected to the stated motivation. It is hard to connect the stated need for causal reasoning to overcome misleading context and the exact operations carried out by the proposed layer. As such, I believe this paper tells a story that is not quite supported by the method and the experiments. I am open to being corrected as this assessment may be the result of a misunderstanding.

---

> ### Author Response · Authors · 2022-11-12
> **Response to Reviewer UmuF (the first part)**
>
> We thank the reviewer for the feedback. However, we believe that there might be many misunderstandings regarding our work. We have addressed all your concerns point by point in two pages. We would appreciate it if you could take a look and get back to us if there are any further confusions.
>
> ### Q1: Confusion about our motivation.
>
> Our motivation seems to be challenged that context is not always bad. We approve of this opinion as well. In fact, we have mentioned this point at the end of the qualitative results, and we would clarify it hereinbelow.  For tasks where samples are insufficient or targets are obscure, context is indeed helpful. However, even in these scenes (long tail/few shot/weak supervision), intervention on the context is proved effective as is introduced in the related work. It is because, the occurrence of an object is due to its occurrence rather than "it should occur", *i.e.*,  a more robust prediction should come from the object itself rather than the context, and classification by context is more likely the expediency and a suboptimal solution. We admit that our method may fail when the instance is too obscure to be found, while still arguing that forcing the model to seek the instances themselves is generally beneficial. Actually, we also mentioned "develop a self-adapting method to judge if the task needs to mitigate contextual bias" in future work, and we promise to move the detailed explanation here to "weakness and future work" in the new version.
> ### Q2: Confusion in Section 3.
> It seems that the reviewer might not be familiar with the causality in computer vision. In this part, we would clarify the background knowledge as well as our writing in detail.
>
> **Why a misleading relationship between X & Y might be reinforced specifically by attention?** There are two reasons for this argument: (1) The usage of attention is usually not explicitly supervised. Hence, in the training, the attention tends to emphasize whatever it regards helpful for the recognition, and what is emphasized contains either the objects themselves or the background context. Therefore, when the recognition of one class is biased to relying on the context, the attention cannot make an amendment but continue on reinforcing it. (2) The attentions we discuss in this paper are implemented on the feature map extracted from the backbone. Thereby, the attention only attends to how the prediction is obtained by the extracted image contents, *i.e.*, the $X \rightarrow Y$.
>
> **What it means to break the connection between the context and the image in (2c)?** It is a prescriptive rule in Statistics. In causal theory, $do(X=x)$ denotes "forcing all samples of X to be x". Such operations change the data distribution, hence the name of "intervention". By intervention, the value of X is fixed and C can not affect X, thus the edge $C \rightarrow X$ being cut off.
>
> However, the do-operation alters the data distribution, which is prohibitive in reality. As a substitute, we can utilize observed data (without do) to deduce the intervention effect (with do). Recent years have developed several identical equations including the backdoor adjustment and frontdoor adjustment. The backdoor adjustment revealed in Eq. 1 is one of the most famous equations, whose rigorous derivation can be found on Page211 in *The Book of Why*.
>
> **What is the paradigm of applying causality in computer vision?** We feel this problem is beneficial for the misunderstanding despite not being asked. In most works, they first propose a causal graph (SCM) clarifying the causality in their scenes. Then, in allusion to the problems in SCM, they adopt the according solutions, *e.g.*, the causal intervention for the confounder. Finally and most importantly, they implement the solution in their scenes. Take the original backdoor adjustment for instance, what are $P(Y|X,C=c)$ and $P(c)$ is left to be dealt with.

---

> ### Author Response · Authors · 2022-11-12
> **Response to Reviewer UmuF (the second part)**
>
> ### Q3: Why is our method essentially a causal intervention?
> (1) As is mentioned above, the key of causal intervention lies in how to implement the backdoor adjustment. In Sec. 4.2, we spend a whole chapter on further deducing and implementing of backdoor adjustment: Eq. 5 is implemented as the multiple sampling on $x$ and $P(e_k^n)$ in Eq. 7 is implemented as reweighting. We admit this chapter may be abstruse due to many knowledge of Statistics, but we believe every step is reasonable. It is appreciated if any concrete confusion or doubt on any deduction and implementation in Sec. 4.2 is proposed in time. (2) In fact, multi-head operation being viewed as a form of intervention is seen in several works (*e.g.,* Interventional Few-shot Learning). However, we are the first one to present the complete deduction as shown in Sec 4.2. Besides, we prove that it is the general multiple sampling operation that really works, and the reweighting is also important, which is also absent in previous work.
> ### Q4: Confusion about the results.
> **"Modest" improvement**. Our results are challenged that having small improvements over the baseline, but we believe that there are some misunderstandings. In Fig 1b, the proposed results are only half of our method (mere multiple sampling without re-weighting), but still obtaining +2.2% on mAP and +2.9% on mIOU without any extra parameters. In Table 4, our method has +3.4% on mAP, while owning $17\times$ fewer FLOPs increment than the comparable method. In Table 6 (Table 9 in the original paper), we have +3.7% and +4.3% on C-S mIOU and I-S mIOU. For most works in the semantic segmentation, SOTAs usually only have 2-3% improvement over baseline, hence, why are our improvements "modest"? If there is any doubt that IoU scores are still relatively low, please refer to the response Q4 for reviewer JL6E.
>
> **Why the (small) improvement indicates the problem is indeed being successfully addressed?** Firstly, as is argued above, our improvement is not small, especially in the segmentation task. Then, this claim may be too strong, in that it is impossible to completely remove the impact of context; we have only claimed that contextual bias is successfully mitigated. Finally, it is intuitive that when the mIOU is significantly improved, the model can capture a more accurate attention map, and thus rely more on the foreground and less on the background, which is in accord with our purpose and motivation.
>
> We can understand the reviewers' opinion that context is important to correctly identify e.g. a small/low-resolution object. However, as is explained in Q1, context is somewhat a suboptimal solution, and it is at least sackless to focus the network on the learning of foreground instances. In fact, our motivations as well as results are appreciated by all other reviews, and it is of great appreciation if there are any changes of impressions on our paper.

---

> ### Author Response · Authors · 2022-12-05
> **Looking forward to a discussion before the deadline**
>
> Dear Reviewer,
>
> Thanks again for your great efforts in reviewing our paper!
>
> We have addressed all your questions as well as other reviewers' in detail. As the deadline for the discussion is fast approaching, we are really looking forward to having a discussion with you on the OpenReview system. Would you mind checking our response and letting us know if you have further questions?
>
> With sincere regards,
>
> Authors of Paper 1897

---

> ### Author Response · Authors · 2022-12-12
> **Looking forward to a reponse before the deadline**
>
> Dear Reviewer UmuF,
>
> We want to send you a friendly reminder that the second stage of discussion will be completed soon.
>
> It is greatly appreciated if you are willing to reconsider your score based on our responses, and we really want to know whether our responses address your concerns. If there is any other concern that we could not address in the response, please feel free to let us know and we would be happy to provide further explanation.
>
> With sincere regards,
>
> Authors of Paper 1897

---

### Author Response · Authors · 2022-11-20
**Further response to all reviewers**

Thank you all for the timely response. It seems that there exists a common concern about the scope of our paper and why we do not evaluate on the contextual bias dataset. For this question, we propose a detailed clarification. It is appreciated if you have any further feedback on our response.

Thanks to the reminder from reviewer GZYH and JL6E, we are aware that the evaluations on the OOD (or contextual bias) datasets can be a good indicator for debiasing. However, we also argue that the OOD task is not a suitable main task for us, *i.e.,* pursuing the SOTAs on these tasks conflicts with the goal of applying on **general** visual tasks. The reasons are presented below:

**(1)** OOD datasets (*e.g.*, ImageNet-A, NICO and MetaShift) all include the human-annotated context partitions, which are expensive and unavailable in other tasks. Hence, for one thing, methods for other tasks have no designs for utilizing such partitions; for another, if the ground-truth context is not available, most OOD methods have to rely on the pre-clustering context, which is inaccurate and sees obvious regression.

**(2)** The conflicts between IID (Independent Identical Distribution) test and OOD (Out of Distribution) test. Methods performing excellently on OOD data tend to perform badly on IID data. Most datasets in visual tasks are a mixture of IID and OOD data. Although the bottleneck and most of the bad cases may lie in the OOD part, IID data comprises a large part of the test set. In contrast, in the contextually biased datasets mentioned above, the test set is completely composed of OOD data. (The concern is also raised by review UmuF)

**(3)** Problem of multiple instances. In most OOD datasets, the annotations of foreground target and background context are both single-label, *e.g.*, "cat (person)" and "cat (car)". However, the images in the nature are much more complicated, where there are multiple foregrounds and background objects. It is even more tricky when we are required to recognize all of them, where the foreground of one class is the background of another.

**(4)** Conclusively, there have been some dominating paradigms for OOD problems like ERM and IRM, on which most of the OOD methods are based. Despite the SOTA of these methods on OOD datasets, they do not work well for other tasks. Specifically, these methods can effectively resolve the bottlenecks of common datasets, however, they also bring about regression on IID part. Moreover, most of these methods perform particularly unsatisfying on multi-label tasks. As a result, if we are required to pursue the SOTA on these OOD datasets, we can not reach the goal of "applying for general visual tasks".

However, we agree with the reviews that we should include the experiments on OOD datasets to reveal the capacity of contextual debiasing in our method. We have added relative experiments in the paper (see Table 5 and Table 6 in the revised paper). Specifically, we conduct experiments on the OOD datasets ImageNet-A and Metashift as well as a self-sampled multi-label OOD dataset, and our method achieves comparable performance compared to the SOTA. What's more, the responses from JL6E and GZYH both regard that our performance on ImageNet-A (35.3 VS 35.6) is not "convincing due to inferiority to SOTAs", however, as is mentioned above, it is very difficult to achieve the SOTA on every task. Compared to OOD methods that pursue ultimate performance on OOD datasets, our advantage lies in mitigating contextual bias (Table 5, 6) and simultaneously achieving advancements for general visual tasks (Table 1,2,7,8), and thus, the obvious progress on the contextually biased datasets may have demonstrated our capacity of debiasing.

(Auxiliary) Contextual bias has been reported to be the bottleneck for many visual tasks [1,2,3,4,5,6]. The scope of all these papers as well as ours is to alleviate the bias and meanwhile achieve the overall improvements on these tasks. Although they have achieved remarkable performance in their tasks, neither of their frameworks is based on the ERM or IRM, nor of them is evaluated particularly on OOD datasets. It may be a little bit strict to require them to achieve the SOTAs on both original tasks and OOD datasets.

[1] Visual Commonsense R-CNN. CVPR2020

[2] Causal Intervention for Weakly-Supervised Semantic Segmentation. NIPS2020

[3] Interventional Few-Shot Learning. NIPS2020

[4] Counterfactual VQA: A Cause-Effect Look at Language Bias. CVPR2021

[5] Deconfounded Image Captioning: A Causal Retrospect. TPAMI

[6] Contextual debiasing for visual recognition with causal mechanisms. CVPR2022

---

### Decision · Program_Chairs · 2023-01-20

**Decision:**

Accept: poster

**Justification For Why Not Higher Score:**

1. The implementation of the causality is somewhat unclear.
2. Only multilabel classification is demonstrated. The visual attention should be justified in wider range such as semantic segmentation.

**Justification For Why Not Lower Score:**

Overall, realizing the causality in visual recognition is an interesting idea. The proposed method would attract broad interest in the CV community.

**Metareview: Summary, Strengths And Weaknesses:**

This paper presents a novel attention mechanism with causal justifications that remove the spurious contextual correlation for robust visual recognition. Experimental results on MS-COCO and PASCAL-VOC demonstrate that the proposed attention outperforms baselines.

Strength:
1. A well-written presentation of motivation.
2. The proposed attention is in line with the causal justification, which makes it well-motivated.
3. Comprehensive ablations.

Weakness:
1. The robustness of the attention needs more quantitative analysis
2. As raised by Reviewer UmuF, context is indeed a part of the reasoning process of visual recognition. Authors should not claim that context is harmful. In fact, the authors should argue that the "removal of context" per se is a good result of exploiting context.
3. Some implementation details are missing.


**Note From Pc:**

if the above contains the word "oral" or "spotlight" please see: "oral" presentation means -> notable-top-5% and "spotlight" means -> notable-top-25%. As stated in our emails, we are disassociating presentation type from AC recommendations

**Summary Of Ac-Reviewer Meeting:**

Reviewer UmuF rated 3 mainly due to the misunderstandings of the “causal terms” in the paper. I’ve checked the reviewer’s background and he is indeed limited in such background knowledge. The authors provided comprehensive head-to-head and toe-to-toe feedback. However, he is not responsive. In fact, after rebuttal, two reviewers decided to raise the score from negative to positive. Overall, three reviewers are willing to accept the paper.